# A computational model predicts sex-specific responses to calcium channel blockers in mammalian mesenteric vascular smooth muscle

Gonzalo Hernandez-Hernandez[1], Samantha C O'Dwyer[1], Pei-Chi Yang[1], Collin Matsumoto[1], Mindy Tieu[1], Zhihui Fong[1], Timothy J Lewis[2], L Fernando Santana[1]*, Colleen E Clancy[1,3]*

[1]Department of Physiology & Membrane Biology, University of California, Davis, Davis, United States; [2]Department of Mathematics, University of California, Davis, Davis, United States; [3]Center for Precision Medicine and Data Sciences, University of California, Davis, Davis, United States

*For correspondence:
lfsantana@ucdavis.edu (LFS);
ceclancy@ucdavis.edu (CEC)

Competing interest: The authors declare that no competing interests exist.

**Abstract** The function of the smooth muscle cells lining the walls of mammalian systemic arteries and arterioles is to regulate the diameter of the vessels to control blood flow and blood pressure. Here, we describe an in silico model, which we call the 'Hernandez–Hernandez model', of electrical and $Ca^{2+}$ signaling in arterial myocytes based on new experimental data indicating sex-specific differences in male and female arterial myocytes from murine resistance arteries. The model suggests the fundamental ionic mechanisms underlying membrane potential and intracellular $Ca^{2+}$ signaling during the development of myogenic tone in arterial blood vessels. Although experimental data suggest that $K_V1.5$ channel currents have similar amplitudes, kinetics, and voltage dependencies in male and female myocytes, simulations suggest that the $K_V1.5$ current is the dominant current regulating membrane potential in male myocytes. In female cells, which have larger $K_V2.1$ channel expression and longer time constants for activation than male myocytes, predictions from simulated female myocytes suggest that $K_V2.1$ plays a primary role in the control of membrane potential. Over the physiological range of membrane potentials, the gating of a small number of voltage-gated $K^+$ channels and L-type $Ca^{2+}$ channels are predicted to drive sex-specific differences in intracellular $Ca^{2+}$ and excitability. We also show that in an idealized computational model of a vessel, female arterial smooth muscle exhibits heightened sensitivity to commonly used $Ca^{2+}$ channel blockers compared to male. In summary, we present a new model framework to investigate the potential sex-specific impact of antihypertensive drugs.

## eLife assessment

The study is of importance for the cardiac modeling field by developing a novel mathematical model with sex difference. The data are **compelling**, and the model is helpful for mechanistic understanding, and thus is also **important** for experimental physiology. The model is based on experimental data and validated against some experimental data.

## Introduction

Our primary objective was to develop and implement a novel computational model that comprehensively describes the essential mechanisms underlying electrical activity and $Ca^{2+}$ dynamics in arterial

**eLife digest** High blood pressure is a major risk factor for heart disease, which is one of the leading causes of death worldwide. While drugs are available to control blood pressure, male and female patients can respond differently to treatment. However, the biological mechanisms behind this sex difference are not fully understood.

Blood pressure is controlled by cells lining the artery walls called smooth muscle cells which alter the width of blood vessels. On the surface of smooth muscle cells are potassium and calcium channels which control the cell's electrical activity. When calcium ions enter the cell via calcium channels, this generates an electrical signal that causes the smooth muscle to contract and narrow the blood vessel. Potassium ions then flood out of the cell via potassium channels to dampen the rise in electrical activity, causing the muscle to relax and widen the artery.

There are various sub-types of potassium and calcium channels in smooth muscle cells. Here, Hernandez-Hernandez et al. set out to find how these channels differ between male and female mice, and whether these sex differences could alter the response to blood pressure medication.

The team developed a computational model of a smooth muscle cell, incorporating data from laboratory experiments measuring differences in cells isolated from the arteries of male and female mice. The model predicted that the sub-types of potassium and calcium channels in smooth muscle cells varied between males and females, and how the channels impacted electrical activity also differed. For instance, the potassium channel Kv2.1 was found to have a greater role in controlling electrical activity in female mice, and this sex difference impacted blood vessel contraction. The model also predicted that female mice were more sensitive than males to calcium channel blockers, a drug commonly prescribed to treat high blood pressure.

The findings by Hernandez-Hernandez et al. provide new insights into the biological mechanisms underlying sex differences in response to blood pressure medication. They also demonstrate how computational models can be used to predict the effects of drugs on different individuals. In the future, these predictions may help researchers to identify better, more personalized treatments for blood pressure.

---

myocytes. We aimed to uncover the key components necessary and sufficient to fully understand the behavior of arterial vascular smooth muscle myocytes and the cellular response to variations in pressure. The model represents the first-ever integration of sex-specific variations in voltage-gated $K_V2.1$ and $Ca_V1.2$ channels, enabling the prediction of sex-specific disparities in membrane potential and the regulation of $Ca^{2+}$ signaling in smooth muscle cells from systemic arteries. To further investigate sex-specific responses to antihypertensive medications, we extended our investigation to include a one-dimensional (1D) representation of tissue. This approach enabled us to simulate and forecast the effects of $Ca^{2+}$ channel blockers within the controlled environment of an idealized mesenteric vessel. It is worth noting that this computational framework can be expanded to predict the consequences of antihypertensive drugs and other perturbations, transitioning seamlessly from single-cell to tissue-level simulations.

Previous mathematical models (*Jacobsen et al., 2007*; *Kapela et al., 2008*; *Yang et al., 2003*; *Parthimos et al., 1999*) of vascular smooth muscle myocytes generated to describe the membrane potential and $Ca^{2+}$ signaling in vascular smooth muscle cells have described the activation of G-protein-coupled receptors (GPCRs) by endogenous or pharmacological vasoactive agents activating inositol 1,4,5-trisphosphate ($IP_3$) and ryanodine (RyR) receptors, resulting in the initiation of calcium waves. Earlier models have also provided insights into the contraction activation by agonists and the behavior of vasomotion. In a major step forward, the Karlin model (*Karlin, 2015*) incorporated new cell structure data and electrophysiology experimental data in a computational model that predicted the essential behavior of membrane potential and $Ca^{2+}$ signaling arising from intracellular domains found in arterial myocytes. One notable limitation of earlier models is that they are based entirely on data from male animals. Furthermore, many data used to parameterize the Karlin model were obtained from smooth muscle from cerebral arteries. While cerebral arteries are important for brain blood flow, they do not control systemic blood pressure. Furthermore, they do not take into consideration the role of $K_V2.1$ channels in the regulation of smooth muscle cell membrane potential.

The function of the smooth muscle cells that wrap around small arteries is to regulate the diameter of these vessels. Arterial myocytes contract in response to increases in intravascular pressure (*Bayliss, 1902*). Based on work largely done using cerebral arterial smooth muscle, a model has been proposed in which this myogenic response is initiated when membrane stretch activates $Na^+$-permeable canonical TRPC6 (*Welsh et al., 2002*; *Spassova et al., 2006*) and melastatin-type TRPM4 (*Earley et al., 2004b*; *Earley et al., 2007*). A recent study in smooth muscle from mesenteric arteries identified two additional TRP channels to the chain of events that link increases in intravascular pressure to arterial myocyte depolarization: TRPP1 (PKD1) and TRPP2 (PKD2) channels (*Sharif-Naeini et al., 2009*; *Bulley et al., 2018*). Together, these studies point to an elaborate multi-protein complex that plays a critical role in sensing pressure and initiating the myogenic response by inducing membrane depolarization and activating voltage-gated, dihydropyridine-sensitive L-type $Ca_V1.2$ $Ca^{2+}$ channels (*Moosmang et al., 2003*; *Knot and Nelson, 1998a*). $Ca^{2+}$ entry via a single or small cluster of $Ca_V1.2$ channels produces a local increase in intracellular free $Ca^{2+}$ ($[Ca^{2+}]_i$) called a '$Ca_V1.2$ sparklet' (*Navedo and Santana, 2013*; *Navedo et al., 2005*; *Navedo et al., 2006*; *Amberg et al., 2007*). Activation of multiple $Ca_V1.2$ sparklets produces a global increase in $[Ca^{2+}]_i$ that activates myosin light chain kinase, which initiates actin-myosin cross-bridge cycling and thus contraction (*Nelson et al., 1990*).

Negative feedback regulation of membrane depolarization and $Ca^{2+}$ sparklet activation occurs through the activation of large-conductance, $Ca^{2+}$-activated $K^+$ ($BK_{Ca}$) channels as well as voltage-dependent $K_V2.1$ and $K_V1.5/1.2$ $K^+$ channels (*O'Dwyer et al., 2020*; *Amberg and Santana, 2006*; *Nelson et al., 1995*; *Plane et al., 2005*). $BK_{Ca}$ channels are organized into clusters along the sarcolemma of arterial myocytes (*Sato et al., 2019*) and are activated by $Ca^{2+}$ sparks resulting from the simultaneous opening of ryanodine receptors type 2 (RyR2) located in a specialized junctional sarcoplasmic reticulum (SR) (*Nelson et al., 1995*; *Ledoux et al., 2006*; *Brayden and Nelson, 1992*; *Jaggar et al., 1998a*; *Knot et al., 1998b*). Because the input resistance of arterial myocytes is high (*Pucovský and Bolton, 2006*; *Yuan et al., 1993*) (about 2–10 GΩ), even relatively small currents (10–30 pA) produced by the activation of a small cluster (*Nelson et al., 1995*; *Jaggar et al., 1998b*; *Wang et al., 2004*) of 6–12 $BK_{Ca}$ channels by a $Ca^{2+}$ spark can transiently hyperpolarize the membrane potential of these cells by 10–30 mV. Accordingly, decreases in $BK_{Ca}$, $K_V1.2$, $K_V1.5$, and/or $K_V2.1$ channels depolarize arterial myocytes, increasing $Ca_V1.2$ channel activity, $[Ca^{2+}]_i$, and contraction of arterial smooth muscle (*Amberg and Santana, 2006*; *Zhong et al., 2010*; *Archer et al., 1998*; *Lu et al., 2002*; *Amberg et al., 2004*).

A recent study by *O'Dwyer et al., 2020* suggested that $K_V2.1$ channels have dual conducting and structural roles in mesenteric artery smooth muscle with opposing functional consequences. Conductive $K_V2.1$ channels oppose vasoconstriction by inducing membrane hyperpolarization. Paradoxically, by promoting the structural clustering of the $Ca_V1.2$ channel, $K_V2.1$ enhances $Ca^{2+}$ influx and induces vasoconstriction. Interestingly, $K_V2.1$ protein is expressed to a larger extent in female than in male arterial smooth muscle. This induced larger $Ca_V1.2$ clusters and activity in female than in male arterial myocytes.

Here, we describe a new model, which we call the 'Hernandez–Hernandez model', of mesenteric smooth muscle myocytes that incorporates new electrophysiological and $Ca^{2+}$ signaling data suggesting key sex-specific differences in male and female arterial myocytes. The model simulates membrane currents and their impact on membrane potential as well as local and global $[Ca^{2+}]_i$ signaling in male and female myocytes. The Hernandez–Hernandez model predicts that $K_V2.1$ channels play a critical, unexpectedly large role in the control of membrane potential in female myocytes compared to male myocytes. Importantly, our model predicts that clinically used antihypertensive $Ca_V1.2$ channel blockers cause larger reductions in $Ca_V1.2$ currents in female than in male arterial myocytes.

Finally, we present a one-dimensional (1D) vessel representation of electrotonically coupled arterial myocytes connected in series. Predictions from the idealized vessel suggest that $Ca^{2+}$ channel blockers are more potent in females, resulting in a more substantial $[Ca^{2+}]_i$ reduction in female arterial smooth muscle compared to male. The Hernandez–Hernandez model demonstrates the importance of sex-specific differences in $Ca_V1.2$ and $K_V2.1$ channels and suggests the fundamental electrophysiological and $Ca^{2+}$-linked mechanisms of the myogenic tone. The model also points to testable hypotheses underlying differential sex-based pathogenesis of hypertension and distinct responses to antihypertensive agents.

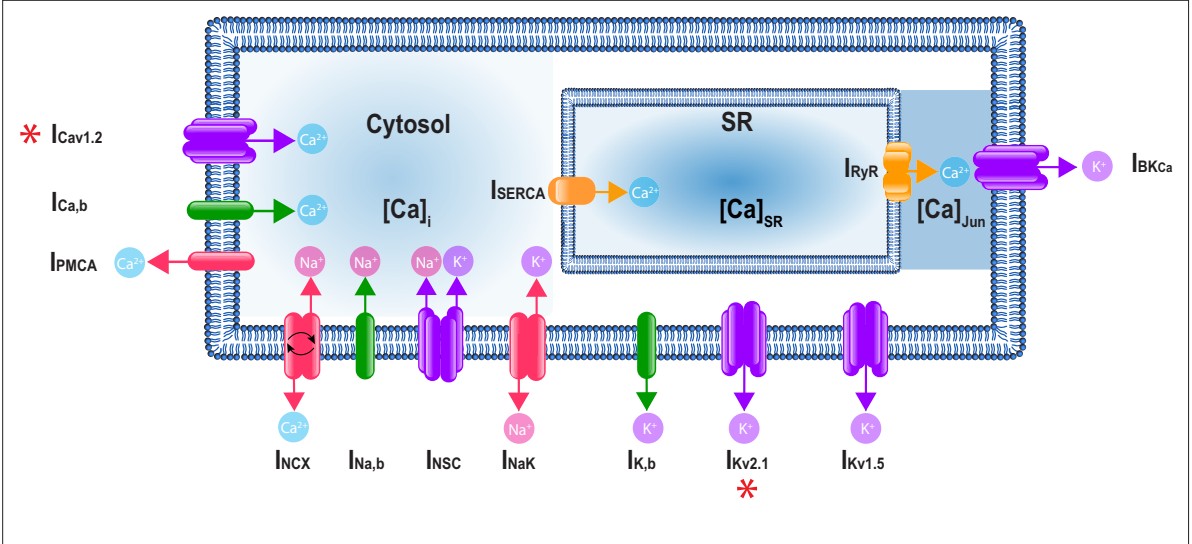

**Figure 1.** A schematic representation of the Hernandez–Hernandez model. The components of the model include major ion channel currents shown in purple including the voltage-gated L-type calcium current ($I_{Ca}$), nonselective cation current ($I_{NSC}$), voltage-gated potassium currents ($I_{Kv1.5}$ and $I_{Kv2.1}$), and the large-conductance $Ca^{2+}$-sensitive potassium current ($I_{BKCa}$). Currents from pumps and transporters are shown in red including the sodium/potassium pump current ($I_{NaK}$), sodium/calcium exchanger current ($I_{NCX}$), and plasma membrane ATPase current ($I_{PMCA}$). Leak currents are indicated in green including the sodium leak current ($I_{Na,b}$), potassium leak current ($I_{K,b}$), and calcium leak current ($I_{Ca,b}$). In addition, two currents in the sarcoplasmic reticulum are shown in orange: the sarcoplasmic reticulum Ca-ATPase current ($I_{SERCA}$) and ryanodine receptor current ($I_{RyR}$). Calcium compartments comprise three discrete regions including cytosol ($[Ca]_i$), sarcoplasmic reticulum ($[Ca]_{SR}$), and the junctional region ($[Ca]_{Jun}$). Red stars (*) indicate measured sex-specific differences in ionic currents.

## Results

In this study, we developed a computational model of the electrical activity of an isolated vascular smooth muscle cell (*Figure 1*). A key goal was to optimize and validate the model with experimental data and then use the model to predict the effects of measured sex-dependent differences in the electrophysiology of smooth muscle myocytes.

In constructing the model, we first set out to measure the kinetics of the voltage-gated L-type $Ca_V1.2$ currents ($I_{Ca}$) in male and female myocytes using $Ca^{2+}$ as the charge carrier as shown in *Figure 2*. These data provided information on the kinetics of $Ca^{2+}$-dependent activation and inactivation of $I_{Ca}$. $I_{Ca}$ is critical in determining cytosolic concentration $[Ca^{2+}]_i$ in vascular mesenteric smooth muscle cells and is the predominant pathway for $Ca^{2+}$ entry (*Moosmang et al., 2003*; *Navedo and Santana, 2013*; *Navedo et al., 2005*; *Amberg et al., 2007*; *Knot et al., 1998b*; *Hill-Eubanks et al., 2011*). Experiments using whole-cell patch-clamp were undertaken to measure the time constants of activation and deactivation (*Figure 2A*) and inactivation (*Figure 2B*) in male and female mesenteric artery smooth muscle cells shown as black and blue symbols, respectively. While the data from male (n = 10) and female (n = 12) myocytes showed comparable activation time constants, there was an observable trend of faster inactivation in the female cells in the lower voltage range, but the differences were not statistically significant. Steady-state activation and inactivation were also measured as shown in *Figure 2C*, with male data in black symbols and female as blue symbols. No observable differences in the gating characteristics of the male and female $I_{Ca}$ were measured. Finally, the current–voltage relationship is shown from measurements in female (blue) and male (black) in *Figure 2D*. This analysis suggests that the amplitude of $I_{Ca}$ was larger in female than in male myocytes over a wide range of membrane potentials.

We next used the experimental measurements to build and optimize a Hodgkin–Huxley model based on the data described above. The model includes voltage-dependent activation and inactivation gating variables, dL and dF, respectively. We modeled both gates following the approach by *Kernik et al., 2019*. It is important to note that smooth muscle cells operate within a voltage regime defined by the window current, which ranges between –45 mV and –20 mV. Under these conditions,

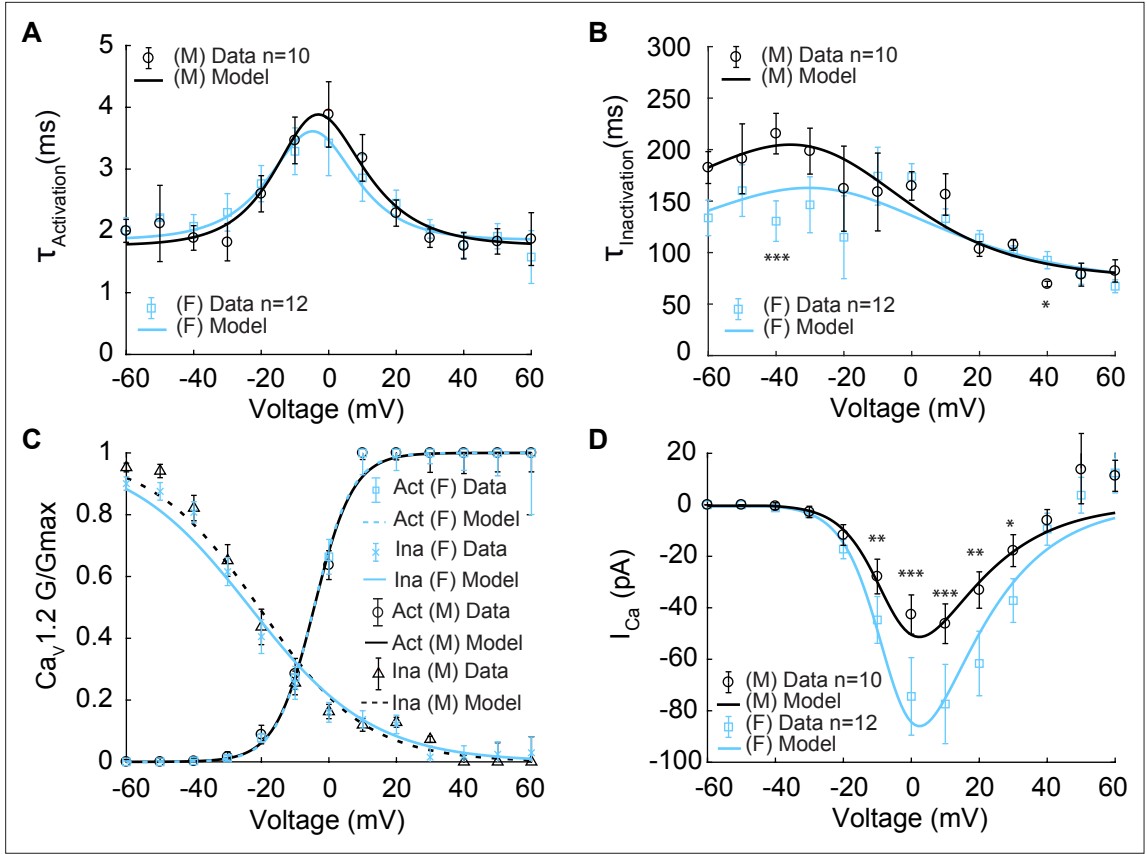

**Figure 2.** Experimentally measured and modeled L-type calcium currents ($I_{Ca}$) from male and female vascular smooth muscle (VSM) cells. Properties of $I_{Ca}$ are derived from measurements in male and female VSM cells isolated from the mouse mesenteric arteries following voltage-clamp steps from –60 to 60 mV in 10 mV steps from a –80 mV holding potential. Experimental data is shown in black circles for male (n=10) and blue squares for female (n=12). Model fits to experimental data are shown with black solid lines for male and blue solid lines for female. (**A**) Male and female time constants of $I_{Ca}$ activation. (**B**) Male and female time constants of $I_{Ca}$ inactivation. (**C**) Male and female voltage-dependent steady-state activation and inactivation of $I_{Ca}$. (**D**) Current–voltage (I–V) relationship of $I_{Ca}$ from male and female VSM myocytes. *p<0.05, **p<0.01, ***p<0.001. Error bars indicate mean ± SEM.

[Ca²⁺]ᵢ remains below 1 µM. Therefore, we did not consider the Ca²⁺-dependent inactivation gating mode of the channel (**Kapela et al., 2008**; **Fleischmann et al., 1994**).

The model of $I_{Ca}$ is described by:

$$I_{Ca} = P_{Ca} * dL * dF * \frac{z_{Ca}^2 F^2 V}{RT} \left( \frac{[Ca]_i e^{\frac{z_{ca}FV}{RT}} - [Ca]_{out}}{e^{\frac{z_{ca}FV}{RT}} - 1} \right) \tag{1}$$

where $P_{Ca}$ is the ion permeability, R is the gas constant, F is the Faraday's constant, and $z_{Ca}$ is the valence of the Ca²⁺ ion. Parameters were optimized to male and female experimental data as shown for activation time constants ($\tau_{activation}$) and inactivation time constants ($\tau_{inactivation}$) as solid lines in **Figure 2A, B**, respectively. Model optimization to male and female activation and inactivation curves are shown in **Figure 2C**. The model was also optimized to the $I_{Ca}$ current–voltage (I–V) relationships shown as solid lines in **Figure 2D**.

We next set out to determine sex differences in voltage-gated K⁺ currents ($I_K$) in male and female mesenteric smooth muscle cells. $I_K$ is produced by the combined activation of $K_V$ and $BK_{Ca}$ channels. Following the approach previously published by **O'Dwyer et al., 2020**, we quantified the contribution of $K_V$ ($I_{KV}$) and $BK_{Ca}$ ($I_{BKCa}$) current to $I_K$. K⁺ currents were recorded before and after the application of the channel blocker iberiotoxin (IBTX; 100 nm). Once identified the contribution $I_{BKCa}$ current, we isolated the voltage-gated potassium currents ($I_{KV}$) whose contributors include the voltage-gated potassium channels $K_V1.5$ and $K_V2.1$. The presumed function of $K_V1.5$ and $K_V2.1$ channels on membrane potential

is to produce delayed rectifier currents to counterbalance the effect of the inward currents (**Nelson et al., 1990**; **O'Dwyer et al., 2020**).

Having isolated $I_{KV}$, $K_V$2.1 currents were identified using the application of the $K_V$2.1 blocker ScTx1 (100 nM). As a result, the remaining ScTx1-insensitive component of the $I_{KV}$ current was attributed to $K_V$1.5 channels. The results are shown in **Figure 3**. Experiments using whole-cell patch-clamp were undertaken to measure the steady-state activation $G/G_{max}$ of the $K_V$2.1current ($I_{Kv2.1}$) as shown in **Figure 3A** in female (blue) and male (black) myocytes and no significant differences were observed. Measurements of time constants of activation (**Figure 3B**) of $I_{Kv2.1}$ in the voltage range of –30 to +40 mV in female (blue, n = 10) and male (black, n = 7) myocytes exhibited significant differences. Notably, activation time constants in male myocytes were smaller than those in female myocytes, corresponding to a faster activation rate in males. The current–voltage relationship of $I_{Kv2.1}$ is shown from measurements in female (blue, n = 20) and male (black, n = 10) myocytes in **Figure 3C**. Significant differences were observed in $I_{Kv2.1}$ at various voltages. In **Figure 3C**, data points without asterisks are not considered significant. Similarly, we measured the steady-state activation of the $K_V$1.5 current ($I_{Kv1.5}$) as shown in **Figure 3D** where male and female experimental data in myocytes are shown with blue and black symbols. Properties of $I_{Kv1.5}$ steady-state activation $G/G_{max}$ show minimal sex-specific differences. The current–voltage relationship of $I_{Kv1.5}$ is shown from measurements in female (blue, n = 10) and male (black, n = 7) myocytes in **Figure 3E**. Finally, the current–voltage relationship of the contribution from $I_{Kv1.5}$ and $I_{Kv2.1}$ to the total voltage-gated current ($I_{KVTOT}$) is shown in **Figure 3F** with male and female data shown with black and blue symbols, respectively. Data points in **Figure 3D–F** without asterisks are not significant. The table in **Figure 3H** summarizes the sex-dependent maximal conductance and the current response at specific voltages of –50, –40, –30, and –20 mV for both $I_{Kv1.5}$ and $I_{Kv2.1}$.

To understand the contribution of each $K^+$ current to the total voltage-gated current ($I_{KVTOT}$) in mesenteric vascular smooth muscle cells, we built and optimized a Hodgkin–Huxley model to the data described above. First, we developed a model to describe the $K_V$2.1 current. The optimized model to $K_V$2.1 experimental data contains only a voltage-dependent activation gating variable ($X_{2.1act}$). Since inactivation time is slow and is well estimated by steady-state (**Yang et al., 2003**), we did not consider its effects in our model. The model of $I_{Kv2.1}$ is described by

$$I_{Kv2.1} = G_{Kv2.1} * X_{Kv2.1act} * (V - E_K) \tag{2}$$

where $G_{K2.1}$ is the maximal conductance of $K_V$2.1 channels and $E_K$ is the Nernst potential for potassium. Parameters were optimized to male and female experimental data as shown for activation curves in **Figure 3A**. Model optimization to male and female time constants of activation ($K_V$2.1 $\tau_{Activation}$) is shown as solid lines in **Figure 3B**. The model was also optimized to the $I_{Kv2.1}$ current–voltage (I–V) relationships shown as solid lines in **Figure 3C**.

Similarly, we developed a model for $K_V$1.5. The model was optimized to the $K_V$1.5 experimental data and contains only a voltage-dependent activation gating variable ($X_{Kv1.5act}$). The model of $I_{Kv1.5}$ is described by

$$I_{KV1.5} = G_{Kv1.5} * X_{Kv1.5act} * (V - E_K) \tag{3}$$

where $G_{K1.5}$ is the maximal conductance of $K_V$1.5 channels and $E_K$ is the Nernst potential for potassium. Parameters were optimized to male and female experimental data as shown for activation curves in **Figure 3D**. The model was also optimized to the $I_{Kv1.5}$ current–voltage (I–V) relationships shown as solid lines in **Figure 3E**. From experiments, we optimized the model to reproduce the overall time traces of $K_V$ currents. The model predicted that male and female myocytes have comparable time constants of activation in $I_{Kv1.5}$ as shown in **Figure 3G**. Finally, the optimized model of the total voltage-gated current ($I_{KVTOT}$) is shown in **Figure 3F**. The total voltage-gated $K^+$ current ($I_{KVTOT}$) is the sum of $I_{KV1.5}$ and $I_{Kv2.1}$ mathematically described as

$$I_{KvTOT} = I_{Kv2.1} + I_{Kv1.5} \tag{4}$$

Notably, the main specific sex-specific differences observed in the total voltage-gated $K^+$ current ($I_{KvTOT}$) is attributable to the sex-specific differences in the current produced by $K_V$2.1 channels.

We next analyzed the contribution of large-conductance calcium-activated potassium ($BK_{Ca}$) channels to vascular smooth muscle cell electrophysiology. $BK_{Ca}$ channels are activated by membrane depolarization or increased $[Ca^{2+}]_i$ and are expressed in the membrane of vascular smooth muscle cells with

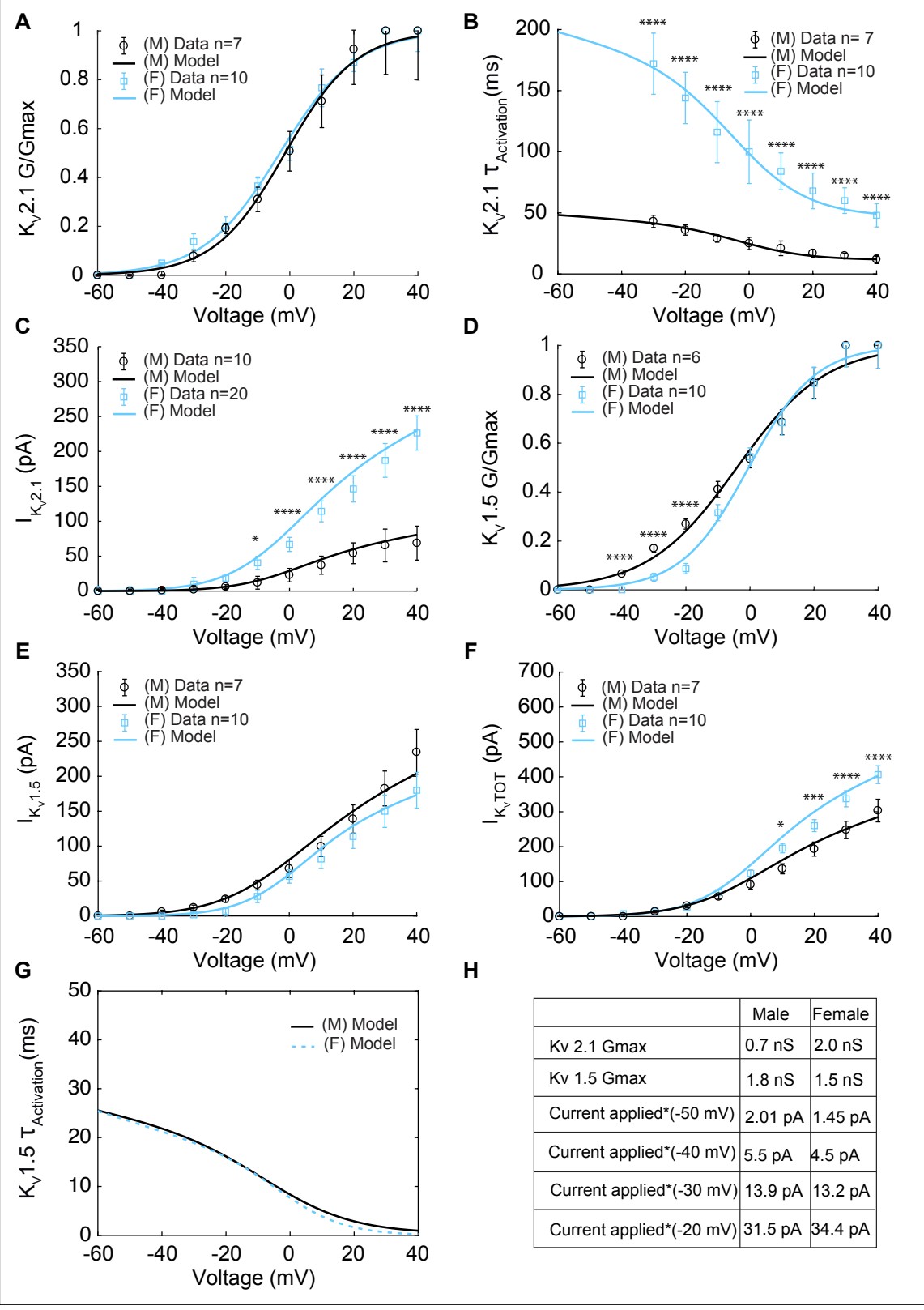

**Figure 3.** Experimentally measured and modeled potassium currents ($I_{KvTOT}$) from male and female vascular smooth muscle cells. The properties of $I_{Kv1.5}$ and $I_{Kv2.1}$ from experimental measurements in male and female vascular smooth muscle cells isolated from the mouse mesenteric arteries were recorded in response to voltage-clamp from –60 to 40 mV in 10 mV steps (holding potential –80 mV). Experimental data is shown as black circles for male and blue squares for female. Model fits to experimental data are shown with black solid lines for male and blue solid lines for female. (**A**) Male (n=7) and

*Figure 3 continued on next page*

*Figure 3 continued*

female (n=10) voltage-dependent steady-state activation of $I_{Kv2.1}$. (**B**) Male (n=7) and female (n=10) time constants of $I_{Kv2.1}$ activation. (**C**) Current–voltage (I–V) relationship of $I_{Kv2.1}$ from male (n=10) and female (n=20) myocytes. (**D**) Male (n=6) and female (n=10) voltage-dependent steady-state activation of $I_{Kv1.5}$. (**E**) Current–voltage (I–V) relationship of $I_{Kv1.5}$ from male (n=7) and female (n=10) myocytes. (**F**) Male (n=7) and female (n=10) total voltage-gated potassium current $I_{KvTOT} = I_{Kv1.5} + I_{Kv2.1}$. (**G**) Predicted male and female time constants of the $I_{Kv1.5}$ activation gate. (**H**) Table showing sex-specific differences in conductance and steady-state total potassium current–voltage dependence. *p<0.05, **p<0.01, ***p<0.001, ****p<0.0001. Data points without asterisks are not significant. Error bars indicate mean ± SEM.

α and β1 subunits (***Nelson et al., 1995***; ***Bao and Cox, 2005***; ***Brenner et al., 2000***). In smooth muscle cells, Ca²⁺ sparks are the physiological activators of $BK_{Ca}$ channels. We relied on the assumption by ***Tong et al., 2011*** that $BK_{Ca}$ currents ($I_{BKCa}$) are produced by two current subtypes, one consisting of α subunits ($I_{BKα}$) and the other consisting of α and β1 subunits ($I_{BKαβ1}$). Experimental evidence indicates that $BK_{Ca}$ channels with αβ1 subunits form clusters in the plasma membrane in specialized junctional domains formed by the SR and the sarcolemma. $BK_{Ca}$ channels with αβ1 subunits colocalize with ryanodine receptors (RyRs) to in the junctional domains. During a Ca²⁺ spark, [Ca²⁺]ᵢ elevations ranging from 10 to 100 μM activate $BK_{Ca}$ channels (***Pérez et al., 1999***; ***Kaßmann et al., 2019***; ***Hill-Eubanks et al., 2011***; ***Jaggar et al., 2000***; ***Zhuge et al., 2002***). In our model, Ca²⁺ sparks are the physiological activators of $BK_{Ca}$ channels.

The mathematical formulation of the $BK_{Ca}$ with αβ1 current ($I_{BKαβ1}$) was optimized to fit the experimental whole-cell electrophysiological data from ***Bao and Cox, 2005*** obtained at room temperature with a $BK_{Ca}$ channel α subunit clone from mSlo-mbr5 and a β1 subunit clone from bovine expressed in

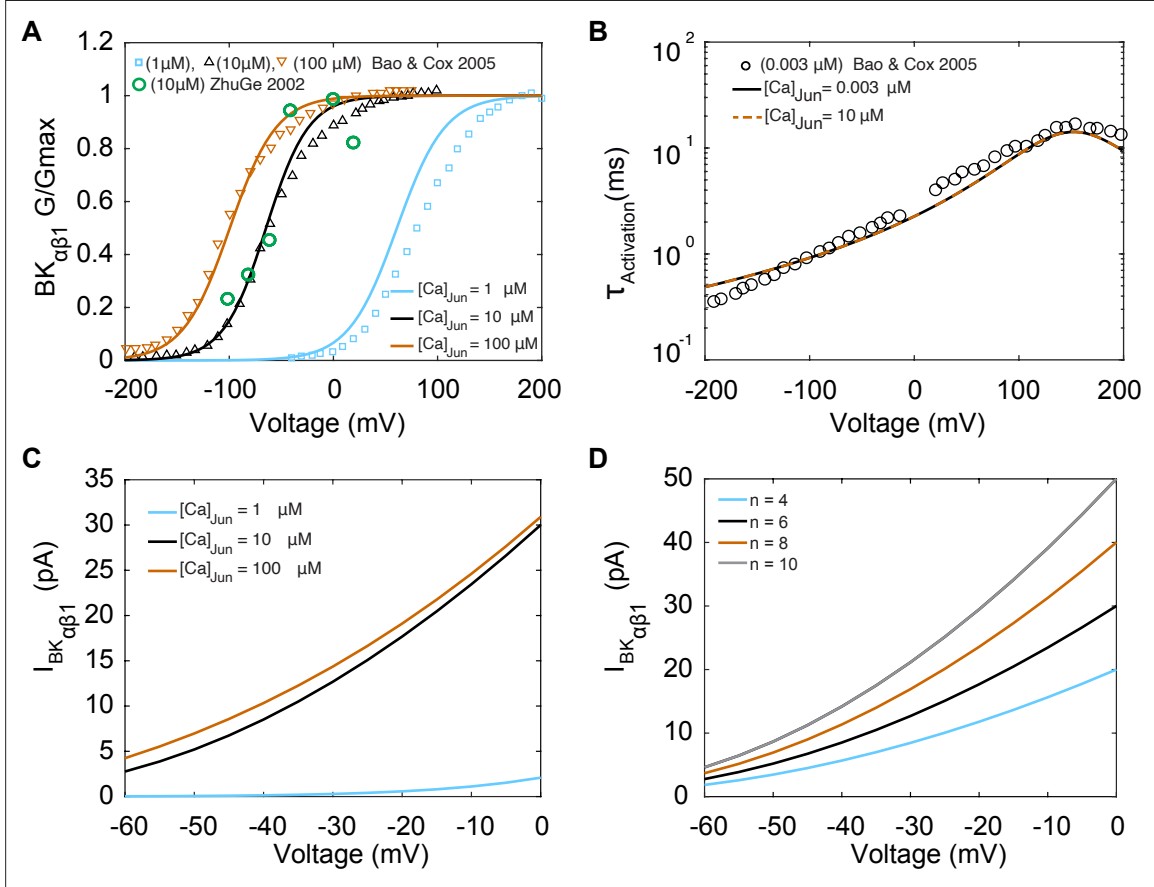

**Figure 4.** Experimentally measured and modeled large-conductance Ca²⁺-activated K⁺ currents ($I_{BKαβ1}$). The model was optimized to data from ***Bao and Cox, 2005***. (**A**) Voltage-dependent activation of $I_{BKαβ1}$ from experiments performed with three different [Ca]$_{Jun}$ concentrations (1 μM, 10 μM, and 100 μM) shown in green circles is the data from ***Zhuge et al., 2002***. (**B**) Voltage-dependent activation time constants with [Ca]$_{Ju}$n = 0.003 μM and simulations [Ca]$_{Ju}$n = 10 μM. (**C**) Simulated I–V curve at different peak levels of [Ca]$_{Jun}$ levels. (**D**) Simulated I–V curve with different $BK_{ca}$ average cluster sizes (N = 4,6, 8, and 10).

*Xenopus laevis* oocytes (**Bao and Cox, 2005**). Experimental data for steady-state activation and time constants of activation are shown in **Figure 4A and B**, respectively. The activation gating variable ($X_{ab}$) depends on both voltage and junctional calcium ($[Ca^{2+}]_{Jun}$). The activation gate was adapted from the Tong–Taggart model (**Tong et al., 2011**). The model of $I_{BK\alpha\beta1}$ is described by

$$I_{BK\alpha\beta1} = P_{BKca} * X_{ab}\left(V, [Ca]_{Jun}\right) * \frac{z_K^2 F^2 V}{RT}\left(\frac{[K]_{in} e^{\frac{z_K FV}{RT}} - [K]_{out}}{e^{\frac{z_K FV}{RT}} - 1}\right) \quad (5)$$

where $P_{BKCa}$ is the $BK_{Ca}$ ion permeability, R is the gas constant, F is Faraday's constant, and $z_K$ is the valance of the potassium ions. Model optimization to activation curves is shown with solid lines in **Figure 4A** at three different $[Ca^{2+}]_{Jun}$ concentrations: 1 μM, 10 μM, and 100 μM. The results from the steady-state activation measurements at 10 μM are also in agreement with the experimental data in vascular myocytes in *Bufo marinus* (**Zhuge et al., 2002**) (green symbols), which suggests that $BK_{Ca}$ channels are exposed to a mean junctional $Ca^{2+}$ concentration ($[Ca^{2+}]_{Jun}$) of 10 μM. Time constants of activation were measured experimentally at $[Ca^{2+}]_{Jun} = 0.003$ μM, and our model was optimized and fit under the same conditions shown in **Figure 4B** as solid lines. Notably when the model was run under predicted $[Ca^{2+}]_{Jun} = 10$ μM conditions as shown in **Figure 4B**, dashed lines, there was no effect of the change in $[Ca^{2+}]_{Jun}$ on the time constant. The predicted current–voltage (I–V) relationships of $I_{BK\alpha\beta1}$ are shown in **Figure 4C** using three different $[Ca^{2+}]_{Jun}$ concentrations: 1 μM, 10 μM, and 100 μM. We observed that the I–V curves are similar at $[Ca^{2+}]_{Jun}$ concentrations of 10 μM (black trace) and 100 μM (orange trace) but markedly reduced when $[Ca^{2+}]_{Jun} = 1$ μM (blue trace). As expected, the amplitude of the current shown in the I–V curves in **Figure 4D** is sensitively dependent on the number of $BK_{Ca}$ channels as shown, and we set $[Ca^{2+}]_{Jun} = 10$ μM and simulated the I–V curves using a $BK_{Ca}$ cluster size of 4, 6, 8 and 10 channels.

In vascular smooth muscle cells, the membrane potential over the physiological range of intravascular pressures is less negative than the equilibrium potential of potassium ($E_K = -84$ mV), suggesting active participation of inward currents regulated by sodium conductance (**Nelson et al., 1990**; **Dwyer et al., 2011**; **Setoguchi et al., 1997**). It has been postulated that basally activating TRP channels generate nonselective cations currents ($I_{NSC}$) that depolarize the membrane potential. We built a model for $I_{NSC}$ as linear and time-independent cation current permeable to $K^+$ and $Na^+$ with permeability ratios $P_{Na}:P_K = 0.9:1.3$ adapted from Tong–Taggart model with a reversal potential ($E_{NSC}$) described by:

$$E_{NSC} = \left(R * \frac{T}{F}\right) * log\left(\frac{P_K * K_{out} + P_{Na} * Na_{out}}{P_K * K_{in} + P_{Na} * Na_{in}}\right); \quad (6)$$

where R is the gas constant, F is the Faraday's constant, T is the temperature, and $Na_{in}$ and $K_{in}$ are the intracellular sodium and potassium intracellular concentrations. Similarly, $Na_{out}$ and $Na_{out}$ are the extracellular sodium and potassium concentrations. The model of $I_{NSC}$ is described by

$$I_{NSC} = I_{NaNSC} + I_{KNSC} \quad (7)$$
$$I_{NaNSC} = G_{NaNSC} * (V - E_{NSC}) \quad (8)$$
$$I_{KNSC} = G_{KNSC} * (V - E_{NSC}) \quad (9)$$

where $I_{NaNSC}$ represents sodium current contribution, $I_{KNSC}$ represents potassium current contribution, and $G_{NaNSC}$ and $G_{KNSC}$ are the maximal conductances of the contributing sodium and potassium currents. In addition, we also included models for leak currents of ion i calculated as

$$I_{i,b} = G_{i,b}(V - E_i) \quad (10)$$

where the Nernst potential of ion i with valance $z_i$ is given by

$$E_i = RT/z_i F \ln([i]_{out}/[i]_{in}), \ i = Na^+, \ K^+ \text{ and } Ca^{2+} \quad (11)$$

where R is the gas constant, F is the Faraday's constant, T is the temperature, and $[i]_{out}$ denotes the extracellular concentration of ion i.

The remaining ionic currents, pumps, and transporters were optimized to data available in the experimental literature and/or taken from computational models of vascular smooth muscle and

cardiac cells. The sodium–potassium pump ($I_{NaK}$) current was modeled using data from smooth muscle cells from mesenteric resistance arteries of the guinea pig (*Tong et al., 2011*; *Nakamura et al., 1999*) and the voltage dependency was adapted from the Luo–Rudy II model (*Luo and Rudy, 1994*). The sodium–calcium exchanger current ($I_{NCX}$) was adapted from the formulation in the ten Tusscher model (*ten Tusscher et al., 2004*) and the Luo–Rudy II model (*Luo and Rudy, 1994*). Finally, the sarcolemma calcium pump ($I_{PMCA}$) current was adapted from the Kargacin model (*Kargacin, 1994*).

$$N_{pow} = \left( Q10^{\frac{T-309.2}{10}} \right); \tag{12}$$

$$N_1 = \frac{\left(K_{out}^{1.1}\right)}{K_{out}^{1.1} + KmNaK_K^{1.1}}; \tag{13}$$

$$N_2 = \frac{\left(Na_{in}^{1.7}\right)}{Na_{in}^{1.7} + KmNaK_{Na}^{1.7}}; \tag{14}$$

$$N_0 = \frac{1.0}{1 + \left(0.1245*exp\left(-0.1*V*\frac{F}{R*T}\right)\right) + \left(2.19e-3*\left(e^{\left(\frac{Na_{out}}{49.71}\right)}\right)*e^{\left(-1.9*V*\frac{F}{R*T}\right)}\right)}; \tag{15}$$

$$I_{NaK} = I_{NaK_{max}} * N_1 * N_2 * N_0 * N_{pow} \tag{16}$$

$$phi_F = exp\left(gammax * V * \frac{F}{R*T}\right) \tag{17}$$

$$phi_R = exp\left((gammax-1)*V*\frac{F}{R*T}\right) \tag{18}$$

$$X_{NCX} = \frac{\left(Na_{in}^3\right)*Ca_{out}*phi_F - \left(Na_{out}^3\right)*Ca_i*phi_R}{1+0.0003*\left(\left(Na_{out}^3\right)Ca_{in}+\left(Na_{in}^3\right)*Ca_{out}\right)}; \tag{19}$$

$$I_{NCX} = P_{NCX} * X_{NCX} \tag{20}$$

$$I_{PMCA} = I_{PMCAbar} * \frac{Ca_i^2}{Ca_i^2 + K_{mPMCA}^2} \tag{21}$$

We next set out to connect the ionic models and models of Ca²⁺ handling to make predictions in the whole cell. In *Figure 5*, experimental data indicate that the electrical activity of isolated mesenteric smooth muscle cells in male and female myocytes recorded in current-clamp mode is characterized by an oscillating membrane potential under physiological conditions. The membrane potential is marked by repetitive spontaneous transient hyperpolarization (TH), a ubiquitous feature of vascular smooth muscle cells (*Jaggar et al., 2000*; *Désilets et al., 1989*; *Bychkov et al., 1997*; *Bae et al., 1999*) as shown in *Figure 5A*. Both male (black trace) and female (blue trace) myocytes exhibited membrane hyperpolarizing transients in the potential range of –50 to –20 mV. Notably, we observed that female myocytes always maintained a higher depolarizing state between the hyperpolarization events compared to male myocytes.

We assessed the predictive capacity of our in silico model by comparing it to experimental data. We first compared the morphology of the membrane potential in experiments *Figure 5A* versus simulations *Figure 5B* in male and female myocytes. Upon comparative analysis between male and female experimental data and simulations, we noted that the baseline membrane potential for male myocytes was around –40 mV, while female myocytes exhibited a slightly more depolarized membrane potential at approximately –30 mV. Despite these variations in baseline membrane potential, both male and female myocytes presented similar peak hyperpolarization values of approximately 10–15 mV, ranging from –50 mV to –30 mV. Similarly, the frequency of THs from multiple myocytes was calculated to be 1–2.8 Hz in the range of –50 mV to –30 mV, which is identical to the simulated frequency.

In the physiological range in which smooth muscle cells operate (−50 to −20 mV), ionic currents are small and produced by the activation of a small number of ion channels. Local fluctuations in the

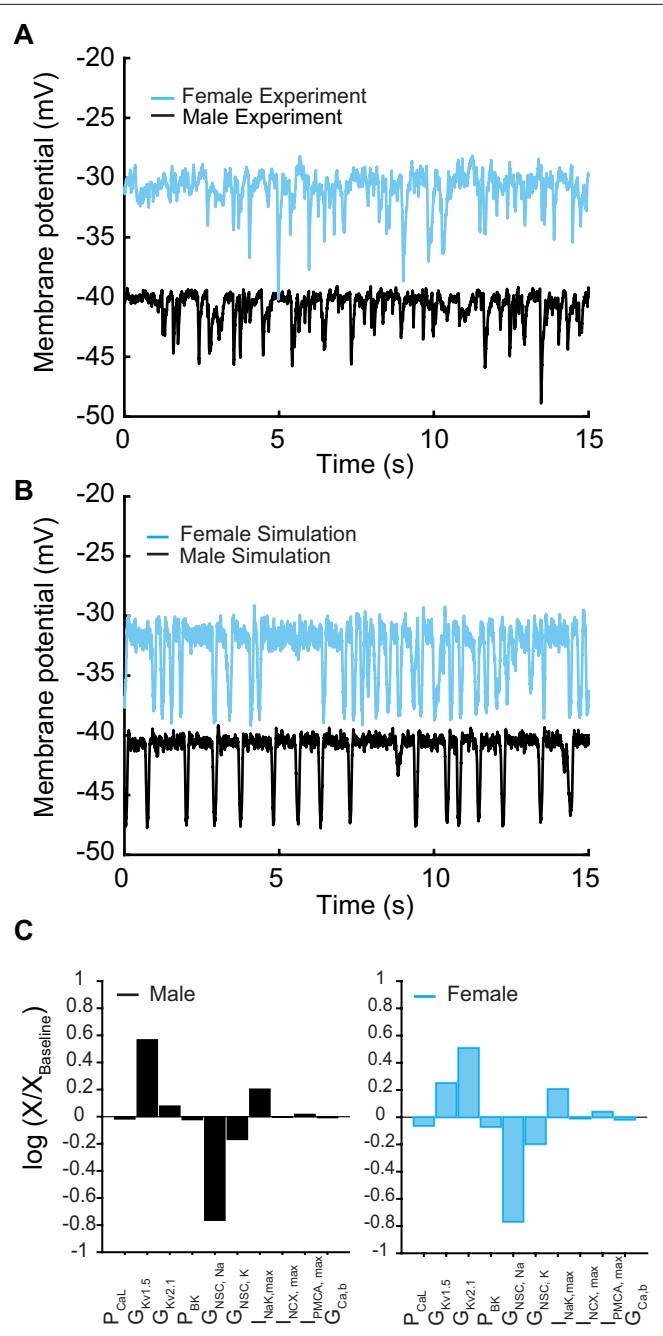

**Figure 5.** Membrane potential from experiments and simulations in male and female vascular smooth muscle myocytes. (**A**) Whole-cell membrane potential recordings in male and female myocytes showing spontaneous repeat transient hyperpolarization of the membrane potential. (**B**) Simulated whole-cell membrane potential with physiological noise. (**C**) Comparison of sensitivity analysis performed around the baseline membrane potential in male and female models using multivariable regression.

function of ion channels lead to noisy macroscopic signals that are important to the variability of vascular smooth muscle cells (***Nelson et al., 1990***). In addition, smooth muscle cells are subject to high input resistance where small perturbations can lead to large changes in the membrane potential (***Nelson et al., 1990***; ***Pucovský and Bolton, 2006***). To approximate the physiological realities, we applied two sources of noise to our deterministic in silico model to simulate the stochastic fluctuations. The first source of the noise was introduced by adding a fluctuating current term to the differential equations describing changes in membrane potential (dV/dt), which represents the combined

effect of the stochastic activity of ion channels in the plasma membrane (*Goldwyn and Shea-Brown, 2011*). Second, we introduced noise into the $[Ca]_{SR}$ to replicate the physiological responses consistent with those observed in experimental studies (*ZhuGe et al., 1999*). Simulated whole-cell membrane potential with physiological noise is shown in *Figure 5B* in male (black trace) and female (blue trace) myocytes.

We conducted a sensitivity analysis to determine which model parameters could underlie the sex-specific differences observed in the experimental data. It is important to note that we have experimental data indicating the amplitude and kinetics for a variety of currents in male and female myocytes. For this reason, those model components were fit to the data, fixed, and were not subject to sensitivity analysis. Our analysis, which focused solely on variations in maximal conductance and maximal ion transport rates of the transmembrane currents, indicated that the nonselective cation currents ($I_{NSC}$) and delayed rectifier currents ($I_{KvTOT} = I_{Kv2.1} + I_{Kv1.5}$) interact to regulate the baseline membrane potential in both male and female vascular smooth muscle myocytes (*Figure 5C*). Given that $I_{KvTOT}$ responds to depolarization, the primary stimulus that triggers depolarization was determined to be attributable solely to the nonselective cation currents ($I_{NSC}$). Indeed, when we adjusted

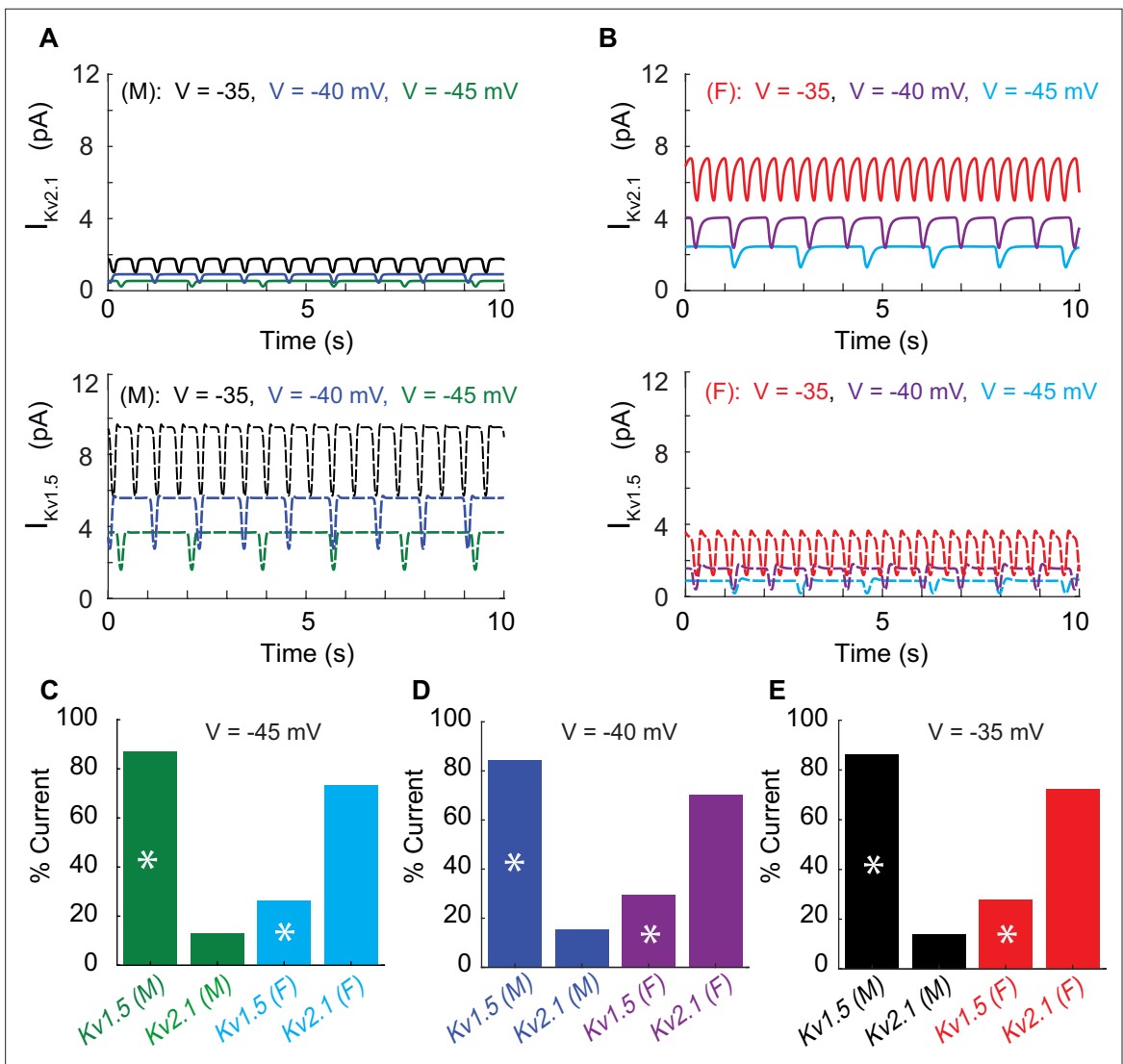

**Figure 6.** Differential effects of voltage-gated potassium current ($I_{KVTOT}$) block in male and female myocytes. (**A**) Simulated time course of male $I_{Kv2.1}$ (top panel, solid traces) and $I_{Kv1.5}$ (lower panel, dashed traces) at three different baseline membrane potentials (–45 mV green, –40 mV blue, and –35 mV black). (**B**) Simulated time course of female $I_{Kv2.1}$ (top panel solid traces) and $I_{Kv1.5}$ (lower panel, dashed traces) at three different baseline membrane potentials (–45 mV light blue, –40 mV purple, and –35 mV red). Current contribution to $I_{KVTOT}$ from $K_v1.5$ (indicated by asterisks) and $K_v2.1$ in male and female myocytes at a baseline membrane potential of –45 mV (**C**), –40 mV (**D**), and –35 mV (**E**).

the conductance of the nonselective cation currents and implemented an increase in the conductance of $I_{NSC}$ in the female model, we readily reproduced the sex-specific baseline membrane potential observed experimentally (**Figure 5A**).

Next, using the whole-cell vascular smooth muscle myocyte computational model, we investigated the sex-specific differences in the contribution to total voltage-gated current ($I_{KVTOT}$) in mesenteric vascular smooth muscle cells. An interesting prediction from the in silico simulations is that at different depolarizing states (−45, −40, and −35 mV) induced by changing the conductance of nonselective cationic leak currents ($I_{NSC}$), the contribution of $I_{Kv2.1}$ and $I_{Kv1.5}$ to $I_{KvTOT}$ is different based on sex. In male vascular myocytes, the contribution to total voltage-gated current ($I_{KVTOT}$) is largely attributable to the current produced by $K_V1.5$ channels as shown in the lower panel in **Figure 6A**. Our results are consistent with previous studies (**Lu et al., 2002**; **Sung et al., 2013**; **Bratz et al., 2005**) in animal rodent male models showing the characteristic behavior of $I_{Kv1.5}$ to control membrane potential. However, the model predicts that in female myocytes the contribution to total voltage-gated current ($I_{KVTOT}$) is largely provided by the current produced by $K_V2.1$ channels as shown in the upper panel in **Figure 6B**. To illustrate this point quantitatively, at a membrane potential of −40 mV, the contribution of $I_{KVTOT}$ from $I_{Kv1.5}$ and $I_{Kv2.1}$ is 86 and 14%, respectively, in male myocytes compared to female myocytes in which the contribution from $I_{Kv1.5}$ and $I_{Kv2.1}$ is 23 and 77%, respectively. Regardless of the depolarization state at −45, −40, or −35 mV, the profiles for male and female myocytes remain essentially the same as shown in **Figure 6C–E**. The in silico simulations suggest a distinctive sex-based function of $K_V1.5$ and $K_V2.1$

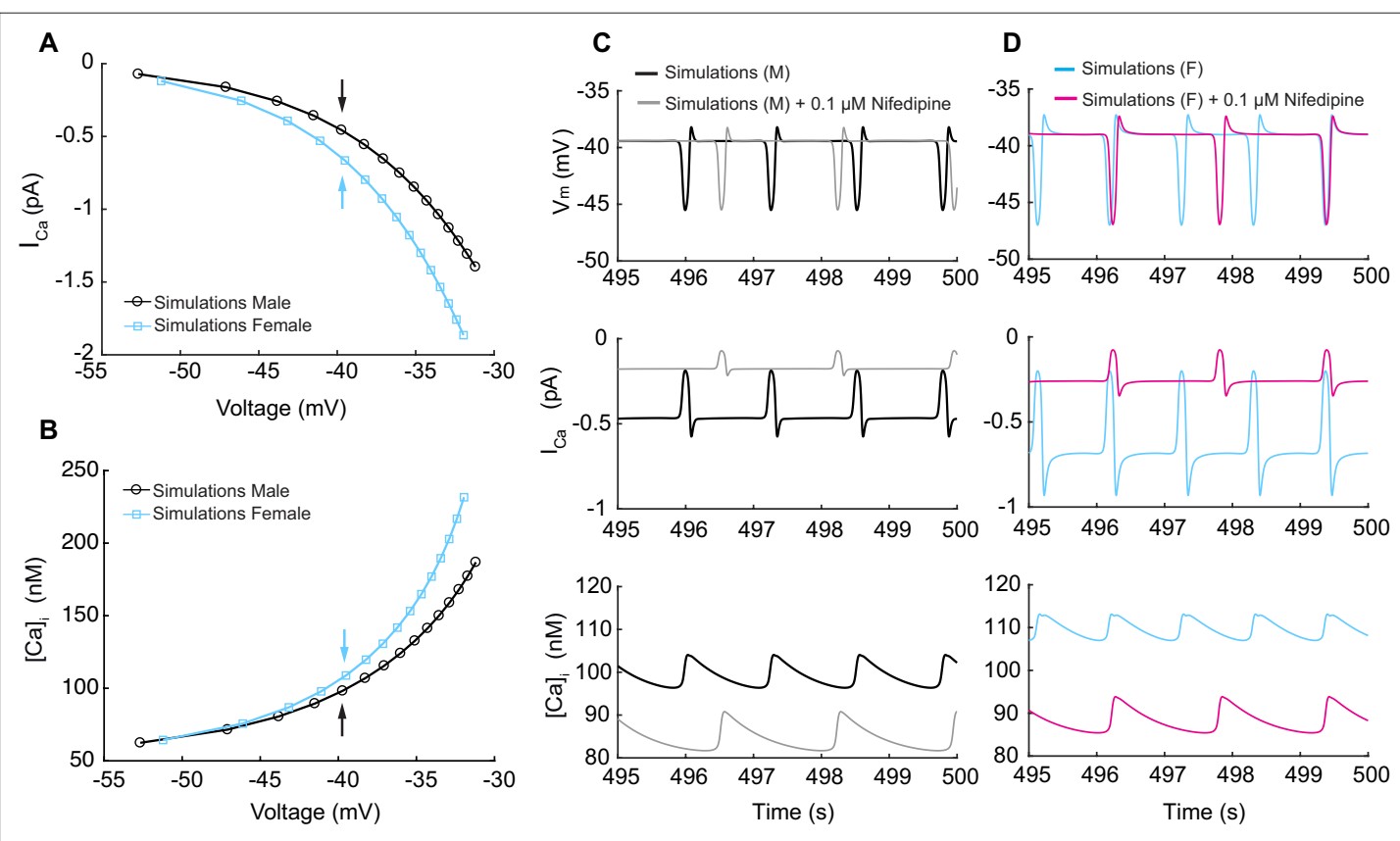

**Figure 7.** Simulated L-type calcium currents ($I_{Ca}$) and calcium influx in male and female vascular smooth muscle cells. (**A**) Male and female whole-cell $I_{Ca}$ membrane potential relationship. (**B**) Male and female intracellular calcium concentration in the cytosolic compartment at indicated membrane potential. (**C**) Time course of membrane potential in male vascular smooth muscle cells before (black) and after (gray) simulated nifedipine application (top panel). Corresponding time course of L-type calcium current $I_{Ca}$ before (black) and after (gray) simulated nifedipine application (middle panel) and intracellular calcium [Ca²⁺]ᵢ concentration before (black) and after (gray) simulated nifedipine application (lower panel). (**D**) Time course of membrane potential in female vascular smooth muscle cells before (blue) and after (pink) simulated nifedipine application (top panel). Corresponding time course of L-type calcium current $I_{Ca}$ before (blue) and after (pink) simulated nifedipine application (middle panel) and intracellular calcium [Ca²⁺]ᵢ concentration before (blue) and after (pink) simulated nifedipine application (lower panel).

channels that produce the delayed rectifier currents to counterbalance the effect of inward currents causing graded membrane potential depolarizations.

Having explored the regulation of graded membrane potential by the activation of $I_{KVTOT}$ to counterbalance the nonselective cations currents ($I_{NSC}$), we next explored the effects of steady membrane depolarization in the in silico vascular smooth muscle cell myocyte model on $I_{Ca}$ in male and female myocytes. We predicted $I_{Ca}$ in our male and female simulations at steady-state membrane depolarization after simulation for 500 s. We observed that as the membrane depolarizes from –55 to –35 mV, $I_{Ca}$ in male myocytes increased from 0 to 1.0 pA while in female myocytes $I_{Ca}$ increased from 0 to 1.5 pA as shown in *Figure 7A*, suggesting that $I_{Ca}$ is larger in female compared to those of male myocytes. We recorded the predicted $[Ca^{2+}]_i$ and observed that $I_{Ca}$ led to a higher calcium influx in female compared to male simulations as shown in *Figure 7B*. To illustrate in detail, we show in *Figure 7C and D* time traces of in silico predictions of membrane voltage at –40 mV (top panel), $I_{Ca}$ (middle panel), and $[Ca^{2+}]_i$ (lower panel) corresponding to the male and female data points indicated by black and blue arrows, respectively, shown in *Figure 7A and B*. In the male case (*Figure 7C*), at a steady membrane potential of –40 mV, L-type calcium $Ca_V1.2$ channels produced a current of 0.5 pA. However, in female simulations (*Figure 7D*), we observed that at a steady membrane potential of –40 mV, L-type calcium $Ca_V1.2$ channels produced a current of 0.65 pA. We calculated that at –40 mV, two $Ca_V1.2$ channels are needed to sustain 0.5 pA of current in male myocytes while three $Ca_V1.2$ channels are needed to sustain 0.65 pA of current in female myocytes. Although the sex-specific differences in male and female simulations at –40 mV are small, a 15 nM difference in the overall response of $[Ca^{2+}]_i$ can have a profound effect on the constriction state of the myocytes. The predictions from the Hernandez–Hernandez model provide a comprehensive picture of physiological conditions and support the idea that a small number of $Ca_V1.2$ channels supply the steady $Ca^{2+}$ influx needed to support a maintained constricted state in small arteries and arterioles (*Fleischmann et al., 1994*; *Rubart et al., 1996*). The differences between males and females are notable in the context of observations indicating varied sex-based responses to antihypertensive agents that target the $Ca^{2+}$ handling system in vascular smooth muscle cells.

Next, we simulated the effects of calcium channel blocker nifedipine on $I_{Ca}$ at a steady membrane potential of –40 mV in male and female simulations. Briefly, previous studies *Davis et al., 1992* have shown that at the therapeutic dose of nifedipine (i.e., about 0.1 μM) L-type Cav1.2 channel currents are reduced by about 60–70%. Accordingly, we decreased $I_{Ca}$ in our mathematical simulations by the same extent. In *Figure 7C and D*, we show the predicted male (gray) and female (pink) time course of membrane voltage at –40 mV (*top panel*), $I_{Ca}$ (middle panel), and $[Ca^{2+}]_i$ (lower panel). First, we observed that in both male and females 0.1 μM nifedipine modifies the frequency of oscillation in the membrane potential by causing a reduction in oscillation frequency. Second, both male and female simulations (middle panels) show that 0.1 μM nifedipine caused a reduction of $I_{Ca}$ to levels that are very similar in male and female myocytes following treatment. Consequently, the reduction of $I_{Ca}$ causes both male and female simulations to reach a very similar baseline $[Ca^{2+}]_i$ of about 85 nM (lower panels). As a result, simulations provide evidence supporting the idea that $Ca_V1.2$ channels are the predominant regulators of intracellular $[Ca^{2+}]$ entry in the physiological range from –40 mV to –20 mV. Importantly, these predictions also suggest that clinically relevant concentrations of nifedipine cause larger overall reductions in $Ca^{2+}$ influx in female than in male arterial myocytes.

Thus far, we have shown the development and application of models of vascular smooth muscle myocytes incorporating measured sex-specific differences in currents from male and female isolated cells. Given that hypertension is essentially a consequence of the spatial organization and function of smooth muscle cells (*Mulvany and Aalkjaer, 1990*; *Martinez-Lemus, 2012*), we next expanded our study to include a 1D tissue representation of electrotonically coupled tissue by connecting arterial myocytes in series.

A well-known phenomenon in excitable systems is that electrotonic coupling between cells results in the minimization of individual cellular differences, thereby producing a smoothing effect across the tissue (*Heppner and Plonsey, 1970*; *Rudy and Quan, 1987*; *Henriquez and Plonsey, 1987*). We simulated 400 female or 400 male vascular smooth muscle myocytes and set the gap junctional conductivity to zero to uncouple the simulated cells. As expected, the uncoupled cells in both male and female cases demonstrated the characteristic behavior of arterial myocytes, exhibiting spontaneous

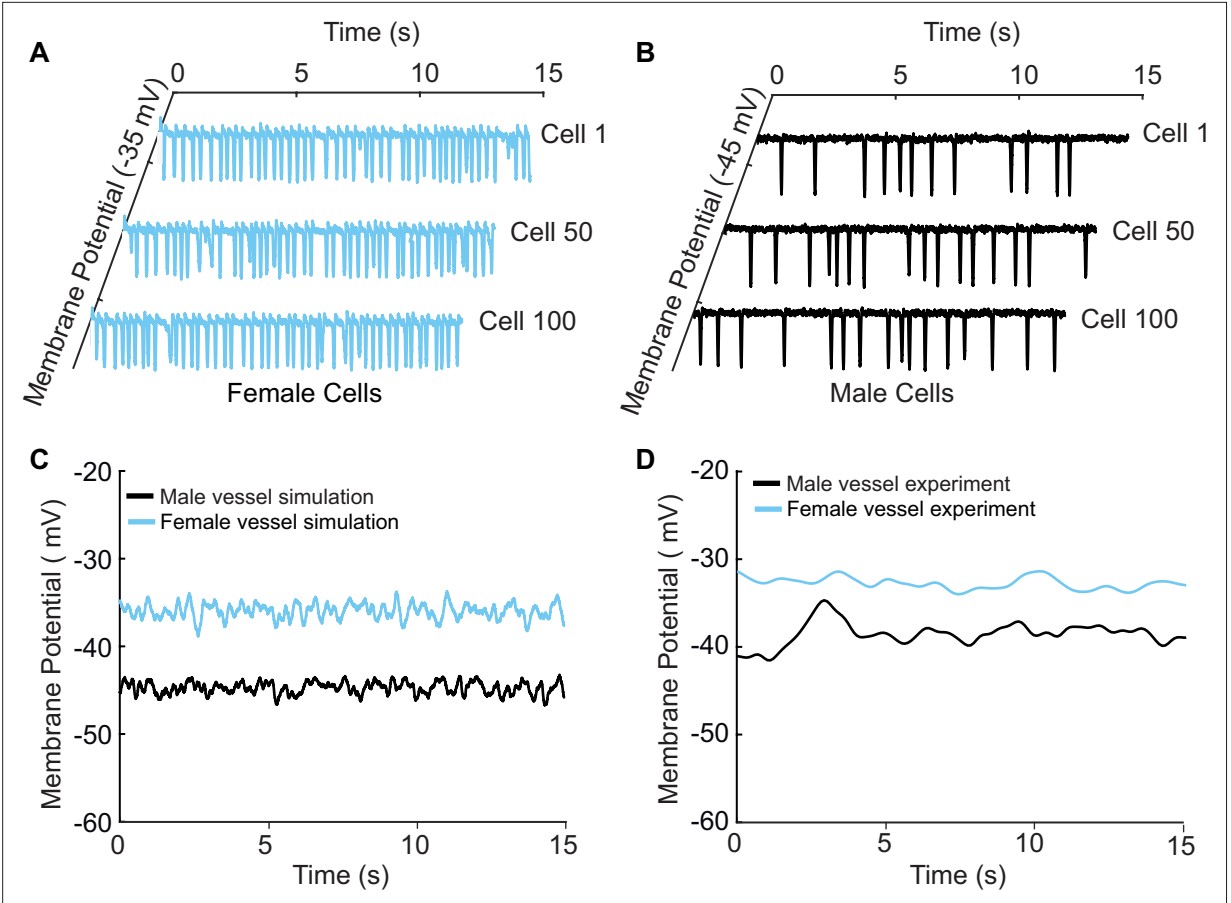

**Figure 8.** A one-dimensional tissue model representation of vascular smooth muscle cells connected in series. (**A**) Uncoupled female vessel simulation showing cell 1, cell 50, and cell 100 at a baseline membrane potential of –35 mV. (**B**) Uncoupled male vessel simulations showing cell 1, cell 50, and cell 100 at a baseline membrane potential of –45 mV. (**C**) Composite female (blue trace) and male (black trace) membrane potential of 400 coupled smooth muscle cells connected with gap junctional resistance of 71.4 Ω cm² in a one-dimensional tissue representation. (**D**) Sharp-electrode records of the membrane potential of smooth muscle in pressurized (80 mmHg) female and male arteries from *O'Dwyer et al., 2020*.

hyperpolarization. Of the 400 cells, we show the simulated representative traces of cell 1, cell 50, and cell 100 for female (*Figure 8A*) and male (*Figure 8B*).

Next, we modeled 400 cells but with electrotonic coupling by setting the simulated gap junctional resistance to 71.4 Ω cm² (*Yamamoto et al., 2001*). In this case, we observed that the spontaneous hyperpolarizations, previously observed in the uncoupled cells, diminished when cells were coupled. The overall smoothing effect observed in *Figure 8C* is attributed to the electrotonic coupling and consequential influence of neighboring cells. The electrical response is consistent across the spatial domain for both male (*Figure 8C*, black trace) and female (*Figure 8C*, blue trace) one-dimensional tissue representations. Notably, the model predicts a more depolarized female membrane potential in the 1D tissue representations consistent with experimental measurements as shown in *Figure 8D*.

Having developed an idealized model of a vessel, we set out to validate the model predictions of variable $[Ca^{2+}]_i$ between males and females by comparing the computed calcium signaling in vascular smooth muscle with experimental recordings (*O'Dwyer et al., 2020*). Given that membrane potential predominantly governs calcium influx in vascular smooth muscle (*Knot and Nelson, 1998a*), we varied the conductance of the nonselective cation currents ($I_{NSCC}$) in our simulations. Tuning of $I_{NSCC}$ was performed to replicate the effects of pressure-induced membrane depolarization, which results in activation of the voltage-gated L-type $Ca^{2+}$ channels and increases $[Ca^{2+}]_i$.

Our simulations (lines) are well validated by experimental recordings (symbols) in *Figure 9A*. A distinctive feature from the model prediction, which was validated by experimental recordings, is the observation that female (*Figure 9A*, blue trace and symbols) vessels accommodate more $[Ca^{2+}]_i$

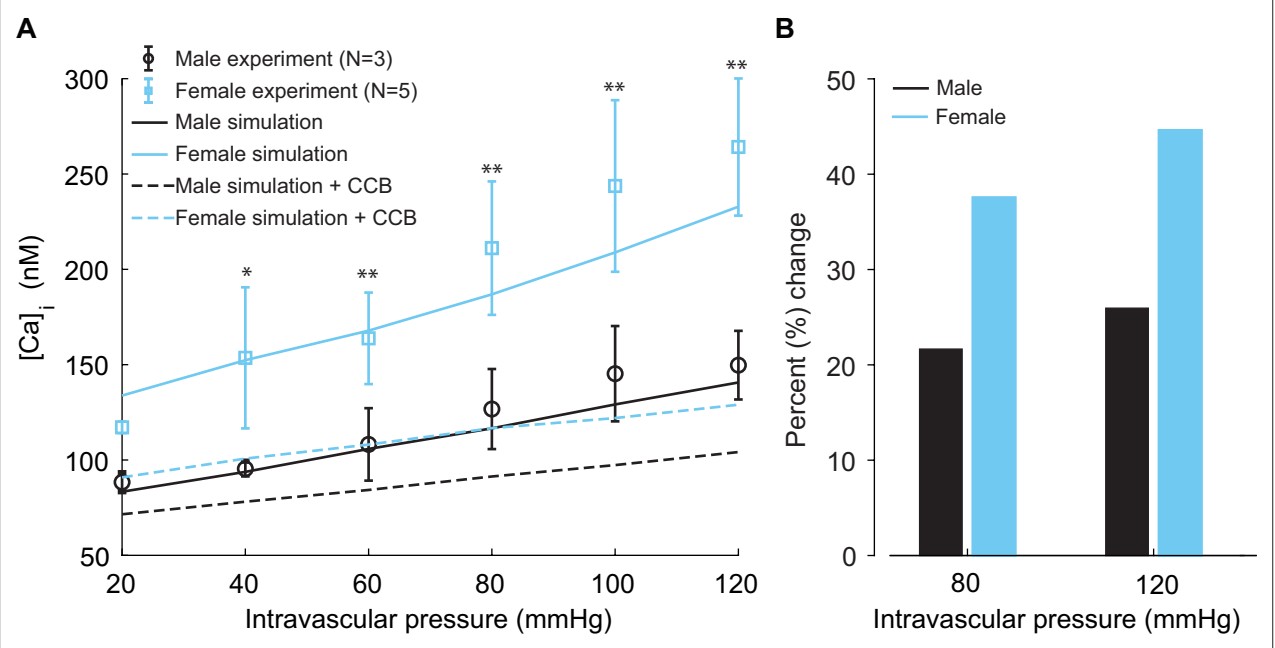

**Figure 9.** Experimentally measured and modeled intracellular calcium [Ca]$_i$ in male and female vessels and response to clinically used L-type Ca$^{2+}$ channel blocker. (**A**) Intracellular calcium [Ca]$_i$ in female (blue symbols, n=3) and male (black symbols, n=5) arteries at intravascular pressures ranging from 20 to 120 mmHg. Simulations showing [Ca]$_i$ in the idealized female and male vessels are shown with blue and black solid lines, respectively. Simulated [Ca]$_i$ after the application of clinically used L-type Ca$^{2+}$ channel blocker nifedipine is shown with dashed lines for male (black) and female (blue). (**B**) Comparison of the percentage change of [Ca]$_i$ in male (black) and female (blue) after the application L-type Ca$^{2+}$ channel blocker nifedipine at 80 mmHg and 120 mmHg. *p<0.05, **p<0.01.

compared to male (*Figure 9A*, black trace and symbols) vessels. Intriguingly, the mechanism of different [Ca$^{2+}$]$_i$ in male and female vessels was revealed in single-cell simulations, which showed attributable sex-based differences in L-type Ca$^{2+}$ currents.

Finally, in our simulations, we computed the effects of [Ca$^{2+}$]$_i$ after the application of clinically relevant calcium channel blocker nifedipine. We observed a substantial reduction of [Ca$^{2+}$]$_i$ in both male (*Figure 9A*, dashed black line) and female (*Figure 9A*, dashed blue line). Significant differences were found in the physiological range of intravascular pressure from 40 to 120 mmHg. In the summary data (*Figure 9B*), we quantified the relative change of [Ca$^{2+}$]$_i$ in male (*black*) and female (*blue*) after the application of 0.1 µM L-type Ca$^{2+}$ channel blocker nifedipine at 80 mmHg and 120 mmHg. Our results show that nifedipine, when applied to male vessels, decreases [Ca$^{2+}$]$_i$ by 22 and 25% at 80 mmHg and 120 mmHg, respectively. However, the same dose of nifedipine when applied to female vessels decreases [Ca$^{2+}$]$_i$ by 38 and 45% at 80 mmHg and 120 mmHg. The results suggest that female arterial smooth muscle is more sensitive to clinically used Ca$^{2+}$ channel blockers than male smooth muscle.

## Discussion

Here, we describe the development, validation, and application of an in silico model to simulate and understand the mechanisms of electrical activity and Ca$^{2+}$ dynamics in a single mesenteric vascular smooth muscle cell. The Hernandez–Hernandez model is the first model to incorporate sex-specific differences in voltage-gated K$_V$2.1 and Ca$_V$1.2 channels and predicts sex-specific differences in membrane potential and Ca$^{2+}$ signaling regulation in the smooth muscle of both sexes from systemic arteries. In the pursuit of stratifying sex-specific responses to antihypertensive drugs, we expanded our exploration to encompass a 1D tissue representation. Such an approach allowed us to simulate and predict the impact of Ca$^{2+}$ channel blockers within a mesenteric vessel. Notably, the computational framework can be expanded to forecast the impact of antihypertensives and other perturbations from single-cell to tissue-level simulations.

To specifically investigate the impact of sex-specific differences measured from ion channel experiments and their impact on membrane potential and $[Ca^{2+}]_i$, we focused on the isolated myocyte in the absence of complex signaling pathways. We first explored the effects of $Ca_V1.2$ and $K_V2.1$ channels on membrane potential as experimental data suggest key sex-specific differences in channel expression and kinetics. Notably, the peak of the current–voltage (I–V) relationship of L-type $Ca_V1.2$ current is 40% smaller in male compared to female myocytes (*Figure 2D*).

Similarly, the peak current–voltage (I–V) relationship of the voltage-gated $K_V2.1$ current ($I_{Kv2.1}$) is 70% smaller in male compared to female myocytes at +40 mV (*Figure 3C*). *O'Dwyer et al., 2020* showed sex-dependent expression of $K_V2.1$ in the plasma membrane, where male arterial myocytes have a total of about 75,000 channels compared to 183,000 channels in female myocytes. Notably, less than 0.01% of channels are conducting in male and female myocytes. In the computational model, we found that to reproduce the experimentally measured amplitude of the $K_V2.1$ I–V curve (*Figure 3C*), a maximum of ~44 male $K_V2.1$ channels was sufficient to reproduce the peak current (68.8 pA at 40 mV). In contrast, ~143 channels were predicted to be needed in female myocytes to reproduce the experimentally measured peak current (226.42 pA at +40 mV) of the $K_V2.1$ I–V relationship. Modeling and simulation led to the prediction that in male arterial myocytes $I_{KvTOT}$ is largely dictated by $K_V1.5$ channels. In contrast, in female arterial myocytes, $K_V2.1$ channels dominate $I_{KvTOT}$ (*Figure 3F*).

An important aspect of the Hernandez–Hernandez model is that it includes $Ca^{2+}$-mediated signaling between RyRs in the junctional SR and $BK_{Ca}$ channel clusters in the nearby sarcolemma membrane. This section of the model is similar to the one included in the Karlin model (*Karlin, 2015*) with some modifications. The Karlin model described how subcellular junctional spaces influence membrane potential and $[Ca^{2+}]_i$ in response to intravascular pressure, vasoconstrictors, and vasodilators. In this study, we reduced the complexity of the model representation of subcellular $Ca^{2+}$ signaling spaces to include just three compartments: the cytosol, SR, and the SR-sarcolemma junction. Our model represents on average the behavior of a single junctional SR unit that is functional in a cell at a time. The model uses a deterministic approach but mimics the process of production of $Ca^{2+}$ sparks that activate $BK_{Ca}$ channel clusters (*Pucovský and Bolton, 2006*). We represented the activity of the RyRs in the junctional domain deterministically in the model so that $Ca^{2+}$ spark-$BK_{Ca}$ currents occur at a frequency of about 1 Hz at –40 mV in a space equivalent to 1% of the total cell surface area of the plasma membrane (*Nelson et al., 1995*).

Based on experimental observations, the Hernandez–Hernandez computational model employs three key assumptions: first, $Ca^{2+}$ sparks in the junctional domain are initiated by activation of RyRs, where RyR gating opening probability is correlated with SR load. Second, $Ca^{2+}$ sparks lead to a $[Ca^{2+}]_{Jun}$ increase between 10 and 20 mM to match the amplitude measured in experiments (*Figure 4A*; *Bao and Cox, 2005*; *Zhuge et al., 2002*). Third, activation of $BK_{Ca}$ channels and the resultant current amplitude derives from the experimentally observed spontaneous outward currents (STOCs) in both amplitude and morphology.

Notably, model simulations revealed important mechanisms that may underlie experimental observations in measurements of membrane potential (*Figure 5A*). The model predicts that the mechanism of intrinsic oscillatory behavior in the vascular myocytes results from a delicate balance of currents. Activation of nonselective cation currents ($I_{NSC}$) likely causes membrane depolarization, but the delayed rectifier currents ($I_{KvTOT}$) oppose them, resulting in membrane potential baseline in the physiological range of –45 to –20 mV. Interestingly, the voltage-gated L-type $Ca_V1.2$ currents activation threshold sits within this range at ~–45 mV. Therefore, small increases in $I_{NSC}$ can overwhelm $I_{KvTOT}$ below –20 mV and result in sufficient depolarization to bring the membrane potential to the threshold for activation of $I_{Ca}$. It is important to note that $I_{KvTOT}$ increases sharply upon depolarization from –45 to –20 mV, resulting in tight control of membrane potential and prevention of large transient depolarization resulting from $I_{NSC}$. Activation of L-type $Ca^{2+}$ channels upon depolarization and subsequent $Ca^{2+}$ release within the small volume junction then activates the $BK_{Ca}$ channels, which results in hyperpolarization. Hyperpolarization reignites the oscillatory cascade as an intrinsic resetting mechanism. Since vascular myocytes are subject to substantial noise from the stochastic opening of ion channels in the plasma membrane, and fluctuations in the local junctional domain components, such as the SR load, RyR opening, and $BK_{Ca}$ channel activity, we included noise in the simulation. To simulate the physiological noise in the vascular smooth muscle cell (*Figure 5B*), we added Gaussian noise to the dV/dt equations and $[Ca]_{SR}$.

Female mesenteric artery myocytes are more depolarized than male myocytes at physiological intravascular pressures (*O'Dwyer et al., 2020*). Our model suggests that female myocytes are more depolarized than male myocytes due to larger nonselective cation currents in female compared to male myocytes, most likely due to the activation of Na$^+$-permeable TRP channels. To our knowledge, the only TRP channels found to regulate the membrane potential of mesenteric artery smooth muscle are TRPP1 and TRPP2 (*Sharif-Naeini et al., 2009*; *Bulley et al., 2018*). Future work will have to determine whether TRPC6 (*Spassova et al., 2006*) and TRPM4 (*Earley et al., 2004b*; *Earley et al., 2007*), which have been shown to mediate the myogenic response of cerebral artery smooth muscle and/or other nonselective cation channels also depolarize mesenteric artery smooth muscle (*Earley and Brayden, 2015*).

The Hernandez–Hernandez model predicts that very few channels (based on total current amplitude) are likely to control the baseline fluctuations in membrane potential in the physiological range of –60 to –20 mV. The intrinsic oscillatory properties of the vascular myocyte operating in the low-voltage regime under conditions of high resistance membrane are similar to other types of oscillatory electrical cells including cardiac pacemaker cells.

As shown in (*Figure 6*), the model predicts that, at –40 mV, the amplitude of steady-state $K_V2.1$ currents is about 0.8 pA in male and 3.3 pA in female arterial myocytes, indicating that the contribution of $K_V2.1$ and $K_V1.5$ channels to membrane potential is different in males and females. At –30 mV, it is 2.34 pA and 9.2 pA in male and female myocytes, respectively. Assuming a single-channel current at –40 and –30 mV of 0.7 pA, we calculated that, on average, in male myocytes a single channel is open at –40 mV and three channels are open at any particular time at –30 mV. In female myocytes, 6 channels are predicted to be open at –40 mV, while 13 are predicted to be active at –30 mV.

The Hernandez–Hernandez model also allowed us to calculate the number of $Ca_V1.2$ channels needed to sustain the steady-state concentration of $[Ca^{2+}]_i$ in the physiological range from –60 to –20 mV (*Figure 7*). The model predicts that at –40 mV in mouse male myocytes two channels were required to generate 0.5 pA of steady-state $Ca_V1.2$ current. On the other hand, we found that in female myocytes, three channels were sufficient to generate 0.65 pA of $Ca_V1.2$ current. These data are consistent with the work of *Rubart et al., 1996*, which suggested that steady-state $Ca^{2+}$ currents at –40 mV were likely produced by the opening of two $Ca_V1.2$ channels in rat cerebral artery smooth muscle cells.

The observation that a very small number of the conducting $K_V2.1$ and $Ca_V1.2$ channels are involved in the regulation of membrane potential and $Ca^{2+}$ influx in male and female arterial myocytes at physiological membrane potentials is important for several reasons. First, the analysis suggests that small differences in the number of $K_V2.1$ and $Ca_V1.2$ channels can translate into large, functionally important differences in membrane potential and $[Ca^{2+}]_i$ and hence affect and control myogenic tone under physiological and pathological conditions. Second, the small number of $K_V2.1$ and $Ca_V1.2$ channels gating between –40 and –30 mV likely makes smooth muscle cells more susceptible to stochastic fluctuations in the number and open probabilities of these channels than in cells where a large number of channels regulate membrane excitability and $Ca^{2+}$ influx (e.g., adult ventricular myocyte; *Guarina et al., 2022*). This, at least in part, likely contributes to $Ca^{2+}$ signaling heterogeneity in vascular smooth muscle.

Hypertension fundamentally manifests through the spatial organization of cellular components, particularly evident in the context of the tunica media, and the middle layer of vessels is predominantly constituted of smooth muscle cells, which play a pivotal role in vessel contraction and relaxation (*Mulvany and Aalkjaer, 1990*; *Martinez-Lemus, 2012*). Such intricate biological machinery is imperative in orchestrating the regulation of blood flow and blood pressure. Our approach began with a process of distillation, aiming to shed light on cellular mechanisms within isolated vascular myocytes from small systemic vessels and arterioles, which control blood pressure, of both male and female mice.

Earlier research has confirmed that in mesenteric arteries the pathogenesis leading to hypertension is largely determined by the downregulation of $K_V2.1$ (*Amberg et al., 2004*) and/or $K_V1.5$ (*Sung et al., 2013*; *Bae et al., 2006*) and a concurrent increase in the activity of $Ca_V1.2$ (*Navedo et al., 2010*) channels. Building upon this knowledge, we broadened our study to encompass a 1D tissue model of electrotonically linked tissue, achieved by connecting arterial myocytes in series. The 1D cable model has anatomical relevance because the structures of third- and fourth-order

mesenteric arteries have a singular layer of vascular myocytes encircling the lumen in a cylindrical arrangement. The cable structure is analogous to an 'unrolled' or lateral arrangement of the vessel. Such an approach allowed a conceptual framework to bridge the gap between understanding the combined effects of membrane potential and $[Ca^{2+}]_i$ in isolated cells and in the wider context of small vessels.

For instance, previous studies have proposed that gap junctions enable vessels to function in a way that is analogous to a large capacitor (*Jaggar et al., 2000*; *Diep et al., 2005*). The gap junctions actively filter and transform single-cell electrical activity into sustained responses across the tissue (*Diep et al., 2005*). Recent studies add to this understanding by demonstrating that Connexin 37 (Cx37), a component of these gap junctions, seems to be expressed in the mesenteric arteries (*Earley et al., 2004a*). In our simulations, we showed (*Figure 8A and B*) that indeed uncoupled cells exhibit a spontaneous oscillatory behavior, which studies have confirmed is not an artifact due to isolation from the vessel but rather an intrinsic behavior required to sustain electrical signals. When the cells are connected (*Figure 8C*), the spontaneous hyperpolarization previously observed in the uncoupled cells diminished, and the effect is attributed to the electrotonic coupling and consequential influence of neighboring cells. In addition, in our simulations, we found that it is required to have stochastic fluctuations to allow the system to average the membrane potential behavior that dictates the amount of $[Ca^{2+}]_i$ in the vessels.

Regarding $Ca_V1.2$ channels, simulations forecast the clinically relevant concentrations (0.1 μM) at which common $Ca^{2+}$ channel blockers (e.g., nifedipine) effectively block $Ca_V1.2$ channels in both male and female smooth muscle (*Figure 9*). Our simulations in isolated arterial myocytes and in the 1D tissue model suggest heightened sensitivity to calcium channel blockers in the female compared to male.

The model predictions are aligned with documented sex-specific differences in antihypertensive drug responses (*Ueno and Sato, 2012*; *Kloner et al., 1996*). Previous studies, notably by Kloner et al., have illustrated this point quantitatively, highlighting a more pronounced diastolic BP response in women (91.4%) compared to men (83%) when treated with dihydropyridine-type channel blockers, such as amlodipine. Importantly, this distinction persisted even after adjusting for confounding factors such as baseline BP, age, weight, and dosage per kilogram (*Kloner et al., 1996*). Another interesting observation from Kajiwara et al. emphasizes that vasodilation-related adverse symptoms occur more frequently in younger women (<50 y) compared to their male counterparts, again suggesting a heightened sensitivity to dihydropyridine-type calcium channel blockers (*Kajiwara et al., 2014*).

## Limitations

The model presented here describes the necessary and sufficient ion channels, pumps, and transporters to describe the electrical activity and $Ca^{2+}$ signaling of an isolated mesenteric smooth muscle cell in the absence of complex signaling pathways. Such an approach enabled us to perform a component dissection to analyze the sex-specific differences observed in the fundamental electrophysiology of male and female myocytes. However, it is well known that vascular smooth muscle cells are subject to a plethora of stimuli from endothelial cells, neurotransmitters, endocrine, and paracrine signals (*Karlin, 2015*). The next phase of the project includes an expansion of the model to incorporate receptor-mediated signaling pathways that are essential for blood pressure control.

Excitation–contraction coupling refers to an electrical stimulus that drives the release of calcium from the SR and results in the physical translocation of fibers that underlie muscle contraction. In the present model, we did not explicitly consider the mechanical description of muscle contraction. Nevertheless, we can imply contractile effects by tracking membrane potential and the elevation of $[Ca^{2+}]_i$ as a proxy.

To conclude, we developed and present the Hernandez–Hernandez model of male and female isolated mesenteric vascular myocytes. An additional limitation of our study is the reliance on predominantly murine data. Although mouse arteries do present numerous parallels with human arteries, including analogous intravascular pressure-myogenic tone relationships, resting membrane potentials, and the expression of typical ionic channels like $Ca_V1.2$, BKCa channels, and RyRs (*Yang et al., 2013*; *Nystoriak et al., 2017*; *Nieves-Cintrón et al., 2017*). Future research should assess the direct applicability and implication of our findings in human subjects.

## Conclusions

The Hernandez–Hernandez model of the isolated mesenteric vascular myocyte was informed and validated with experimental data from male and female vascular myocytes. We then used the model to reveal sex-specific mechanisms of $K_V2.1$ and $Ca_V1.2$ channels in controlling membrane potential and $Ca^{2+}$ dynamics. In doing so, we predicted that very few channels are needed to contribute to and sustain the oscillatory behavior of the membrane potential and calcium signaling. We expanded our computational framework to include a 1D tissue representation, providing a basis for simulating the effect of drug effects within a vessel. The model predictions suggested differences in the response of male and female myocytes to drugs and the underlying mechanisms for those differences. These predictions may constitute the first step toward better hypertensive therapy for males and females.

# Materials and methods

## Experimental

### Animals

This study was performed in strict accordance with the recommendations in the Guide for the Care and Use of Laboratory Animals of the National Institutes of Health. All of the animals were handled according to approved institutional animal care and use committee (IACUC) protocols of the University of California Davis. IACUC protocol number is 22503. In this study, 8- to 12-week-old male and female mice C57BL/6J (The Jackson Laboratory, Sacramento, CA) were used. Animals were housed under standard light-dark cycles and allowed to feed and drink ad libitum. Animals were euthanized with a single lethal dose of sodium pentobarbital (250 mg/kg) intraperitoneally. All experiments were conducted in accordance with the University of California Institutional Animal Care and Use Committee guidelines.

### Isolation of arterial myocytes from systemic resistance arterioles

Third- and fourth-order mesenteric arteries were carefully cleaned of surrounding adipose and connective tissues, dissected, and held in ice-cold dissecting solution ($Mg^{2+}$-PSS; 5 mM KCl, 140 mM NaCl, 2 mM $MgCl_2$, 10 mM glucose, and 10 mM HEPES adjusted to pH 7.4 with NaOH). Arteries were first placed in dissecting solution supplemented with 1.23 mg/ml papain (Worthington Biochemical, Lakewood, NJ) and 1 mg/ml DTT for 14 min at 37°C. This was followed by a second 5 min incubation in dissecting solution supplemented with 1.6 mg/ml collagenase H (Sigma-Aldrich, St. Louis, MO), 0.5 mg/ml elastase (Worthington Biochemical), and 1 mg/ml trypsin inhibitor from *Glycine max* (Sigma-Aldrich) at 37°C. Arteries were rinsed three times with dissection solution and single cells were obtained by gentle trituration with a wide-bore glass pipette. Myocytes were maintained at 4°C in dissecting solution until used.

### Patch-clamp electrophysiology

All electrophysiological recordings were acquired at room temperature (22–25°C) with an Axopatch 200B amplifier and Digidata 1440 digitizer (Molecular Devices, Sunnyvale, CA). Borosilicate patch pipettes were pulled and polished to resistances of 3–6 MΩ for all experiments using a micropipette puller (model P-97, Sutter Instruments, Novato, CA).

Voltage-gated $Ca^{2+}$ currents ($I_{Ca}$) were measured using conventional whole-cell voltage-clamp sampled at a frequency of 50 kHz and low-pass filtered at 2 kHz. Arterial myocytes were continuously perfused with 115 mM NaCl, 10 mM TEA-Cl, 0.5 mM $MgCl_2$, 5.5 mM glucose, 5 mM CsCl, 20 mM $CaCl_2$, and 10 mM HEPES, adjusted to pH 7.4 with CsOH. Micropipettes were filled with an internal solution containing 20 mM CsCl, 87 mM aspartic acid, 1 mM $MgCl_2$, 10 mM HEPES, 5 mM MgATP, and 10 mM EGTA adjusted to pH 7.2 using CsOH. Current–voltage relationships were obtained by exposing cells to a series of 300 ms depolarizing pulses from a holding potential of –70 mV to test potentials ranging from –70 to +60 mV. A voltage error of 9.4 mV due to the liquid junction potential of the recording solutions was corrected offline. Voltage dependence of $Ca^{2+}$ channel activation ($G/G_{max}$) was obtained from the resultant currents by converting them to conductance via the equation $G = I_{Ca}/(\text{test potential} - \text{reversal potential of } I_{Ca})$; normalized $G/G_{max}$ was plotted as a function of test

potential. Time constants of activation and inactivation of $I_{Ca}$ were fitted with a single exponential function.

$I_{Kv}$ recordings were performed in the whole-cell configuration with myocytes exposed to an external solution containing 130 mM NaCl, 5 mM KCl, 3 mM MgCl$_2$, 10 mM Glucose, and 10 mM HEPES adjusted to 7.4 using NaOH. The internal pipette solution constituted of 87 mM K-aspartate, 20 mM KCl, 1 mM CaCl$_2$, 1 mM MgCl$_2$, 5 mM MgATP, 10 mM EGTA, and 10 mM HEPES adjusted to 7.2 by KOH. A resultant liquid junction potential of 12.7 mV from these solutions was corrected offline. To obtain current–voltage relationships, cells were subjected to a series of 500 ms test pulses increasing from –70 to +70 mV. To isolate the different K$^+$ channels attributed to composite $I_K$, cells were first bathed in external $I_K$ solution, subsequently exposed to 100 nM iberiotoxin (Alomone, Jerusalem, Israel) to eliminate any BK$_{Ca}$ channel activity and finally immersed in an external solution containing both 100 nM iberiotoxin and 100 nM stromatoxin (Alomone) to block both BK$_{Ca}$ and K$_V$2.1 activity. Ionic current was converted to conductance via the equation G = I(V-E$_K$). E$_K$ was calculated to be –78 mV. Activation time constants for K$_V$2.1 currents were obtained by fitting the rising phase of these currents with a single exponential function.

BK$_{ca}$-mediated STOCs and membrane potential were recorded using the perforated whole-cell configuration. To measure both, myocytes were continuously exposed to a bath solution consisting of 130 mM NaCl, 5 mM KCl, 2 mM CaCl$_2$, 1 mM MgCl$_2$, 10 mM glucose, and 10 mM HEPES, pH adjusted to 7.4 with NaOH. Pipettes were filled with an internal solution containing 110 mM K-aspartate, 30 mM KCl, 10 mM NaCl, 1 mM MgCl$_2$, 0.5 mM EGTA, and 10 mM HEPES adjusted to a pH of 7.3 with KOH. The internal solution was supplemented with 250 µg/ml amphotericin B (Sigma, St. Louis, MO). STOCs were measured in the voltage-clamp mode and were analyzed with the threshold detection algorithm in Clampfit 10 (Axon Instruments, Inc). Membrane potential was measured using the current-clamp mode.

STOCs were recorded using the perforated whole-cell configuration. The composition of the external bath solution consisted of 134 mM NaCl, 6 mM KCl, 1 mM MgCl$_2$,2 mM CaCl$_2$, 10 mM glucose, and 10 mM HEPES adjusted to a pH of 7.4 using NaOH. Pipettes were filled with an internal solution of 110 mM K-aspartate, 10 mM NaCL, 30 mM KCl, 1 mM MgCl$_2$, 160 µg/ml amphotericin B, and 10 mM HEPES using NaOH to adjust to pH to 7.2. Myocytes were sustained at a holding potential of –70 mV before being exposed to a 400 ms ramp protocol from –140 to +60 mV. A voltage error of 12.8 mV resulting from the liquid junction potential was corrected for offline. K$_{ir}$ channels were blocked using 100 µM Ba$^{2+}$.

## Statistics

Data are expressed as mean ± SEM. All data sets were tested for normality. Normally distributed data were analyzed using $t$-tests or ANOVA. ANOVAs were followed by multiple comparison tests (i.e., Tukey). $p < 0.05$ was considered statistically significant.

## Computational modeling and simulation

### Cell size and structure

The mean capacitance of the cells was experimentally calculated to be 16 ± 3 pF based on all the male and female WT mesenteric C57BL/6J cells utilized in the experiments (N = 45). Assuming the cells are roughly cylindrical in shape, the expected radius should be 2.485 µM and a length of 100 µM, leading to a surface area of $1.6 \times 10^{-5}$ cm$^2$ and a total volume of approximately $1.94 \times 10^{-12}$ l. The cell capacitance of excitable membranes is assumed to be $1.0 \times 10^{-6}$ F/cm$^2$; with the calculated surface area, the estimated total cell capacitance is C$_m$ = 16 pF.

Because the total cell volume is roughly $2 \times 10^{-12}$ l, it is assumed that 50% of the total cell volume is occupied by organelles. There are three main compartments in vascular myocytes important to the regulation of membrane potential and calcium signaling: the cytosol, SR, and specialized junctional domains formed by the SR and the plasma membrane. The cytosol occupies approximately 50% of total cell volume (V$_{cyt}$ = $1.0 \times 10^{-12}$ l). The SR occupies approximately 5% of cell cytosolic volume (V$_{SR}$ = $5.0 \times 10^{-14}$ l), and the junctional domain volume is approximately 1% of the cytosol volume (V$_{Jun}$ = $0.5 \times 10^{-14}$ l) (*Nelson et al., 1995*; *Pérez et al., 2001*; *Pérez et al., 1999*; *Kaßmann et al., 2019*).

## Model development

The male and female in silico models are single whole-cell models based on the electrophysiology of isolated mesenteric vascular smooth muscle myocytes. A schematic of the proposed model is shown in *Figure 1*. The membrane electrophysiology can be described by the differential equation

$$\frac{dV}{dt} = \frac{-I_{ion}}{C_m} \tag{22}$$

where V is voltage, t is time, $C_m$ is membrane capacitance, and $I_{ion}$ is the sum of transmembrane currents. The contribution of each transmembrane current to the total transmembrane ionic current can be described by the following equation:

$$I_{ion} = (I_{Kv1.5} + I_{Kv2.1} + I_{BKCa} + I_{K,b} + I_{Cav1.2} + I_{PMCA} + I_{Ca,b} + I_{NCX} + I_{NSC} + I_{NaK} + I_{Na,b}) \tag{23}$$

The 11 transmembrane currents are generated by ion channels, pumps, and transporters. Currents from ion channels include the voltage-gated L-type calcium current ($I_{Ca}$), the nonselective cation current ($I_{NSC}$), voltage-gated potassium currents ($I_{Kv1.5}$ and $I_{Kv2.1}$), and the large-conductance $Ca^{2+}$-sensitive potassium current ($I_{BKCa}$). Additionally, there are three background or leak currents ($I_{K,b}$, $I_{Ca,b}$, and $I_{Na,b}$). Currents from pumps and transporters include the sodium–potassium pump current ($I_{NaK}$), plasma membrane Ca-ATPase transport current ($I_{PMCA}$), and sodium–calcium exchanger current ($I_{NCX}$).

Cytosolic concentrations of sodium and potassium as a function of time are determined by considering the sum of their respective fluxes into the cytosol.

$$\frac{d\left[K^+\right]_{cyt}}{dt} = -\frac{\left(I_{Kv2.1} + I_{Kv1.5} + I_{BKca} + I_{K,b} - 2I_{NaK} + I_{NSC-K}\right)}{z_K Vol_{cyt} F} \tag{24}$$

$$\frac{d\left[Na^+\right]_{cyt}}{dt} = -\frac{\left(3I_{NCX} + 3I_{NaK} + I_{Na,b} + I_{NSC-Na}\right)}{z_{Na} Vol_{cyt} F} \tag{25}$$

where F is the Faraday's constant, $Vol_{cyt}$ is the cytoplasmic volume, and $z_K$ and $z_{Na}$ are the valence of potassium and sodium ions, respectively.

The calcium dynamics is compartmentalized into three distinct regions: cytosol $[Ca^{2+}]_i$, the SR $[Ca^{2+}]_{SR}$, and the junctional region $[Ca^{2+}]_{jun}$. The cytosol includes a calcium buffer, which we assume can be described as a first-order dynamics process.

### Cytosolic calcium region ($[Ca^{2+}]_i$)

Calcium concentration in this region varies between 100 and 300 nM (*Moosmang et al., 2003*) and is mainly influenced by the following fluxes: transmembrane pumps and transporters, the SR Ca-ATPase ($J_{SERCA}$), diffusion from the junctional domain region ($J_{Jun-Cyt}$) and the calcium buffer calmodulin ($BUF_{CAM}$).

$$\frac{d\left[Ca^{2+}\right]_i}{dt} = -\frac{\left(I_{NCX} + I_{Cav1.2} + I_{Ca,b} - 2I_{PMCA}\right)}{z_{Ca} F V_{Cyt}} - J_{SERCA} + J_{jun-cyt} - \left(k_{BUF_{on}} * Ca_i * \left(BUF_T - BUF_{CAM}\right) - k_{BUF_{off}} * BUF_{CAM}\right) \tag{26}$$

### Sarcoplasmic reticulum region ($[Ca^{2+}]_{SR}$)

Calcium concentration in this region varies between 100 and 150 µM (*ZhuGe et al., 1999*) and is mainly influenced by the SR Ca-ATPase ($J_{SERCA}$) and the flux from the ryanodine receptors ($J_{RyR}$).

$$\frac{d\left[Ca\right]_{SR}}{dt} = \left[\frac{Vol_{Cyt}}{Vol_{SR}}\right] J_{SERCA} - \left[\frac{Vol_{Cyt}}{Vol_{SR}}\right] \left[J_{Ryr}\right] \tag{27}$$

### Junctional region ($[Ca^{2+}]_{jun}$)

Calcium concentration in this region varies between 10 and 100 µM (*Pérez et al., 2001*; *Pérez et al., 1999*) and is mainly influenced by the flux from the ryanodine receptors ($J_{RyR}$) and the diffusion from the junctional region to the cytoplasm ($J_{Jun-Cyt}$)

$$\frac{d[Ca]_{jun}}{dt} = \left[\frac{Vol_{Cyt}}{Vol_{jun}}\right] J_{Ryr} - \left[\frac{Vol_{Cyt}}{Vol_{jun}}\right] J_{jun-cyt} \tag{28}$$

In the model, the flux of $J_{SERCA}$ was adapted from the Luo–Rudy II model (*Luo and Rudy, 1994*), and the flux of $J_{RyR}$ was adapted from previous models of ryanodine receptors activation, originally introduced in the field of cardiac electrophysiology (*Hernandez-Hernandez et al., 2015*; *Greene and Shiferaw, 2021*; *Shiferaw et al., 2003*).

## Parameter optimization and reformulation of the gating ion channel models

The ionic current models of $I_{Ca}$, $I_{Kv2.1}$, and $I_{Kv1.5}$ were optimized using the approach employed by *Kernik et al., 2019*. Here, the open probability $P_o$ of each voltage-dependent gating variable 'n' was defined by opening- and closing-rate voltage-dependent functions $\alpha_n$ and $\beta_n$, respectively, and was modeled by simple exponentials of the form:

$$\alpha_n(V) = x_1 e^{\left(\frac{V}{x_2}\right)} \tag{29}$$

$$\beta_n(V) = x_3 e^{\left(\frac{V}{x_4}\right)} \tag{30}$$

$$\tau_n(V) = \frac{1}{\alpha_n(V) + \beta_n(V)} + x_5 \tag{31}$$

The steady-state availability remains the same as the classical Hodgkin–Huxley formulations, and the time constant values follow a modified version formulation by accommodating an extra parameter $x_5$ in *Equation 31*. ($x_1$, $x_2$, $x_3$, $x_4$, $x_5$) are parameters to be optimized using experimental data. We used the parameter optimization employed by *Kernik et al., 2019*, which minimizes the error between model and experimental data using the Nelder–Mead minimization of the error function. Random small perturbations (<10%) were applied to find local minima to improve data fit. The parameter fit with the minimal error function value after 1000–10,000 perturbations was used as the optimal model fit to the data.

## Cellular simulations with noise

The simulations encapsulate the cumulative effect of stochastic ion channel activity on cell voltage dynamics through the fluctuating current term, $\xi(t)$, into the membrane potential (dV/dt) equation (*Goldwyn and Shea-Brown, 2011*), as shown in *Equation 32*. Here it is assumed $\xi(t)$ is only a function of time and is implemented as Gaussian white noise (*Tanskanen and Alvarez, 2007*).

$$\frac{dV}{dt} = -\frac{I_{total}(V)}{C_m} + \sigma\xi(t) \tag{32}$$

We use the Euler–Maruyama numerical method for updating *Equations 33 and 34* as follows:

$$V(t + \Delta t) = V(t) - \frac{I(V(t))}{C_m}\Delta t + \sigma * \text{randN} * \sqrt{\Delta t} \tag{33}$$

$$\frac{d[Ca]_{SR}}{dt} = \left[\frac{Vol_{Cyt}}{Vol_{SR}}\right] J_{SERCA} - \left[\frac{Vol_{Cyt}}{Vol_{SR}}\right][J_{Ryr}] + \sigma * \text{randN} * \sqrt{\Delta t} \tag{34}$$

where randN is a random number from a normal distribution (N(0,1)) with mean 0 and variance 1. $\Delta t$ is the time step and $\sigma$ is the 'diffusion coefficient', which represents the amplitude of the noise. The numerical method for updating the voltage was forward Euler.

## 1D simulations

The idealized 1D representation of a vessel was developed by connecting 400 Hernandez–Hernandez model cells in series via simulated resistances to represent gap junctions. For each cell in the cable, the Hernandez–Hernandez model was used to compute ionic currents and concentration changes. The temporal transmembrane fluxes of the Hernandez–Hernandez model are related to the spatial or

current flow by a finite difference approximation of the cable equation (*Shaw and Rudy, 1997*; *Jack et al., 1975*; *Hodgkin et al., 1946*)

$$\left[ C_m \left( \frac{V_i^{(t+1)} - V_i^t}{\Delta t} \right) + I_{ion} + I_{stim} \right] = \frac{a}{4 \left( R_{myo} + \frac{R_g}{\Delta x} \right)} \frac{(V_{i-1}^t - 2V_{i-1}^t + V_{i-1}^t)}{\Delta x \Delta x} \tag{35}$$

where $I_{ion}$ represents the individual membrane ionic current densities (pA/pF) of the Hernandez–Hernandez model, $I_{stim}$ is the stimulus current density (pA/pF) set to zero in our simulation, $a$ is the radius of the fiber (5 μM), $C_m$ is the membrane capacity (pA/pF), $V_{it}$ is membrane potential at segment i and time t, $\Delta x$ is the discretization element (100 μM = 0.01 cm). Where $R_{myo}$ is the myoplasmic resistance ($R_{myo}$ = 150 Ω cm) and $R_g$ is the gap junction resistance ($R_g$ = 71.4 Ω cm$^2$).

## Sensitivity analysis

The baseline models in male and female vascular smooth muscle cells were analyzed through a parameter sensitivity assessment using multivariable linear regression, following the methodology introduced by *Sobie, 2009*. The scope of the sensitivity analysis encompassed variations in the maximal conductance and maximal ion transport rates of the transmembrane currents, including $I_{Kv1.5}$, $I_{Kv2.1}$, $I_{BKCa}$, $I_{K,b}$, $I_{Cav1.2}$, $I_{PMCA}$, $I_{Ca,b}$, $I_{NCX}$, $I_{NSC}$, $I_{NaK}$, and $I_{Na,b}$. All other parameters, notably those defining model kinetics, remained constant at the values established by the foundational model. Scaling factors were randomly selected from a log-normal distribution characterized by a median value of 1 and a standard deviation of 0.1 (*Kernik et al., 2019*).

## Simulation protocols

Code for simulations and analysis was written in C++ and MATLAB 2018a. The single vascular smooth muscle code was run on an Apple Mac Pro machine with two 2.7 GHz 12-Core Intel Xeon processors and an HP ProLiant DL585 G7 server with a 2.7 GHz 28-core AMD Opteron processor. Vessel simulations were implemented in C++ and parallelized using OpenMP. The C++ code was compiled with the Intel ICC compiler, version 18.0.3. Numerical results were visualized using MATLAB R2018a by The MathWorks, Inc. All codes and detailed model equations are available on GitHub (copy archived at *ClancyLabUCD, 2023*).

# Acknowledgements

This work was supported by the National Institutes of Health Common Fund Grant OT2OD026580 (to CEC, LFS), National Heart, Lung, and Blood Institute grants R01HL128537 (CEC, LFS), R01HL152681 (to CEC, LFS). UC Davis Department of Physiology and Membrane Biology Research Partnership Fund (to CEC), as well as UC Davis T32 Predoctoral and Postdoctoral Training in Basic and Translational Cardiovascular Medicine fellowship supported in part by NHLBI Institutional Training Grant T32HL086350 (to CM, GHH).

# Additional information

## Funding

| Funder | Grant reference number | Author |
| --- | --- | --- |
| Common Fund | OT2OD026580 | L Fernando Santana<br>Colleen E Clancy |
| National Heart, Lung, and Blood Institute | R01HL128537 | Colleen E Clancy<br>L Fernando Santana |
| National Heart, Lung, and Blood Institute | R01HL152681 | Colleen E Clancy<br>L Fernando Santana |

| Funder | Grant reference number | Author |
|---|---|---|
| University of California, Davis | Department of Physiology and Membrane Biology Research Partnership Fund | Colleen E Clancy |
| National Institutes of Health | T32HL086350 | Gonzalo Hernandez-Hernandez Collin Matsumoto |
| National Institutes of Health | R01NS114210 | L Fernando Santana |

The funders had no role in study design, data collection and interpretation, or the decision to submit the work for publication.

## Author contributions

Gonzalo Hernandez-Hernandez, Conceptualization, Data curation, Software, Formal analysis, Investigation, Methodology, Writing – original draft, Writing – review and editing; Samantha C O'Dwyer, Collin Matsumoto, Zhihui Fong, Data curation, Formal analysis, Investigation, Methodology; Pei-Chi Yang, Conceptualization, Software, Formal analysis, Visualization, Methodology; Mindy Tieu, Conceptualization, Software, Formal analysis, Validation, Investigation, Methodology; Timothy J Lewis, Conceptualization, Resources, Software, Formal analysis, Supervision, Writing – original draft, Project administration, Writing – review and editing; L Fernando Santana, Conceptualization, Resources, Formal analysis, Supervision, Funding acquisition, Validation, Visualization, Methodology, Writing – original draft, Project administration, Writing – review and editing; Colleen E Clancy, Conceptualization, Resources, Data curation, Supervision, Funding acquisition, Investigation, Visualization, Methodology, Writing – original draft, Writing – review and editing

## Author ORCIDs

Gonzalo Hernandez-Hernandez ⓘ http://orcid.org/0000-0001-6940-9005
Pei-Chi Yang ⓘ https://orcid.org/0000-0002-5753-1131
L Fernando Santana ⓘ http://orcid.org/0000-0002-4297-8029
Colleen E Clancy ⓘ https://orcid.org/0000-0001-6849-4885

## Ethics

This study was performed in strict accordance with the recommendations in the Guide for the Care and Use of Laboratory Animals of the National Institutes of Health. All of the animals were handled according to approved institutional animal care and use committee (IACUC) protocols of the University of California Davis. IACUC protocol number is 22503. 8- to 12-week-old male and female mice C57BL/6J (The Jackson Laboratory, Sacramento, CA) were used in this study. Animals were housed under standard light-dark cycles and allowed to feed and drink ad libitum. Animals were euthanized with a single lethal dose of sodium pentobarbital (250 mg/kg) intraperitoneally. All experiments were conducted in accordance with the University of California Institutional Animal Care and Use Committee guidelines.

Reviewer #1 (Public Review): https://doi.org/10.7554/eLife.90604.3.sa1
Reviewer #2 (Public Review): https://doi.org/10.7554/eLife.90604.3.sa2
Reviewer #3 (Public Review): https://doi.org/10.7554/eLife.90604.3.sa3
Author Response https://doi.org/10.7554/eLife.90604.3.sa4

# Additional files

## Supplementary files
• MDAR checklist

## Data availability

The current manuscript is a computational study, all data generated for this manuscript as well as modelling code is available at https://github.com/ClancyLabUCD/sex-specific-responses-to-calcium-channel-blockers-in-mesenteric-vascular-smooth-muscle (copy archived at *ClancyLabUCD, 2023*).

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
