## [Editor Report · eLife assessment]

The study is of importance for the cardiac modeling field by developing a novel mathematical model with sex difference. The data are **compelling**, and the model is helpful for mechanistic understanding, and thus is also **important** for experimental physiology. The model is based on experimental data and validated against some experimental data.

---

## [Referee Report · Reviewer #1 (Public Review)]

The authors developed computational models that capture the electrical and Ca2+ signaling behavior in mesenteric arterial cells from male and female mice. Sex-specific differences in the L-type calcium channel and two voltage-gated potassium channels were carefully tuned based on experimental measurements. To incorporate the stochasticity of ion channel openings seen in smooth muscle cells under physiological conditions, noise was added to the membrane potential and the sarcoplasmic Ca2+ concentration equations. Finally, the models were assembled into 1D vessel representations and used to investigate the tissue-level electrical response to an L-type calcium channel blocker. This comprehensive computational framework helped provide nuanced insight into arterial myocyte function difficult to achieve through traditional experimental methods and can be further expanded into tissue-level studies that incorporate signaling pathways for blood pressure control.

Throughout the paper, model behavior was both validated by experimental recordings and well supported by previously published data. The main findings from the models suggested that sex-specific differences in membrane potential regulation and Ca2+ handling are attributable to variability in the gating of a small number of voltage-gated potassium channels and L-type calcium channels. This variability contributes to a higher Ca2+ channel blocker sensitivity in female arterial vessels. Overall, the study successfully presented novel sex-specific computational models of mesenteric arterial myocytes and demonstrated their use in drug-testing applications.

---

## [Referee Report · Reviewer #2 (Public Review)]

In this study, Hernandez-Hernandez et al developed a gender-dependent mathematical model of arterial myocytes based on a previous model and new experimental data. The ionic currents of the model and its sex difference were formulated based on patch clamp experimental data, and the model properties were compared with single cell and tissue scale experimental results. This is a study that is of importance for the modeling field as well as for experimental physiology.

---

## [Referee Report · Reviewer #3 (Public Review)]

Summary:

This hybrid experimental/computational study by Hernandez-Hernandez sheds new light on sex-specific differences between male and female arterial myocytes from resistance arteries. The authors conduct careful experiments in isolated myocytes from male and female mice to obtain the data needed to parameterized sex-specific models of two important ionic currents (i.e., those mediated by CaV1.2 and KV2.1). Available experimental data suggest that KV1.5 channel currents from male and female myocytes are similar, but simulations conducted in the novel Hernandez-Hernandez sex-specific models provide a more nuanced view. This gives rise to the first of the authors' three key scientific claims: (1) In males, KV1.5 is the dominant current regulating membrane potential; whereas, in females, KV2.1 plays a primary role in voltage regulation. They further show that this (2) the latter distinction drives drive sex-specific differences in intracellular Ca2+ and cellular excitability. Finally, working with one-dimensional models comprising several copies of the male/female myocyte models linked by resistive junctions, they use simulations to (3) predict that sensitivity of arterial smooth muscle to Ca2+ channel-blocking drugs commonly used to treat hypertension is heightened in female compared to male cells.

In my opinion, the following strengths of the work are particularly notable:

• The Methodology is described in exquisite detail in straightforward language that will be easy to understand for most if not all peer groups working in computational physiology. The authors have deployed standard protocols (e.g., parameter fitting as described by Kernik et al., sensitivity analysis as described by Sobie et al.) and appropriate brief explanations of these techniques are provided. The manoeuvre used to represent stochastic effects on voltage dynamics is particularly clever and something I have not personally encountered before. Collectively, these strengthen the credibility of the model and greatly enrich the manuscript.

• The Results section describes findings that robustly support the three key scientific claims outlined in my Summary. It is evident these experiments were carefully designed and carried out with care and intentionality. Several figures show experimental data side-by-side with outputs from the corresponding models. These are an excellent illustration of the power of the authors' novel sex-specific computational simulation platform.

---

## [Author Response]

The following is the authors’ response to the original reviews.

**Public Reviews:**

**Reviewer #1 (Public Review):**
Summary:The authors developed computational models that capture the electrical and Ca2+ signaling behavior in mesenteric arterial cells from male and female mice. A baseline model was first formulated with eleven transmembrane currents and three calcium compartments. Sex-specific differences in the L-type calcium channel and two voltage-gated potassium channels were then tuned based on experimental measurements. To incorporate the stochastic ion channel openings seen in smooth muscle cells under physiological conditions, noise was added to the membrane potential and the sarcoplasmic Ca2+ concentration equations. Finally, the models were assembled into 1D vessel representations and used to investigate the tissue-level electrical response to an L-type calcium channel blocker.Strengths:A major strength of the paper is that the modeling studies were performed on three different scales: individual ionic currents, whole-cell, and 1D tissue. This comprehensive computational framework can help provide mechanistic insight into arterial myocyte function that might be difficult to achieve through traditional experimental methods.The authors aimed to develop sex-specific computational models of mesenteric arterial myocytes and demonstrate their use in drug-testing applications. Throughout the paper, model behavior was both validated by experimental recordings and supported by previously published data. The main findings from the models suggested that sex-specific differences in membrane potential and Ca2+ handling are attributable to variability in the gating of a small number of voltage-gated potassium channels and L-type calcium channels. This variability contributes to a higher Ca2+ channel blocker sensitivity in female arterial vessels. Overall, the study successfully met the aims of the paper.

Thank you for your insightful review and for recognizing the strengths of our study. We appreciate your encouraging comment regarding our multi-scale approach. Indeed, we believe that by systematically connecting these scales—individual ionic currents, whole-cell, and 1D tissue—we can integrate and reconcile experimental and clinical data. We anticipate that this approach will not only provide mechanistic insights into arterial myocyte function that may not be easy to glean from traditional experimental methods but will also facilitate the translation of this information into the development of therapeutic interventions.

Weaknesses:A main weakness of the paper, as addressed by the authors, is the simplicity of the 1D vessel model; it does not take into account various signaling pathways or interactions with other cell types which could impact smooth muscle electrophysiology.

Thank you for highlighting areas for improvement in our study. The strength of computational modeling lies in its iterative nature, allowing us to introduce and examine variables in a systematic manner. While our current model is simplified and does not contain all details, the modular nature of the build will allow continuous expansion to add the important elements described by the reviewer. We are enthusiastic about progressively enriching the model in subsequent studies, introducing signaling pathways in a step-by-step manner, and ensuring their validation with rigorous experimental data.

Another potential shortcoming is the use of mouse data for optimizing the model, as there could be discrepancies in signaling behavior that limit the translatability to human myocyte predictions.

We appreciate this important comment. Our model was parametrized using data from mouse mesenteric artery smooth muscle cells as initial proof of concept. Mouse arteries are a good representation of human arteries, as they have similar intravascular pressure-myogenic tone relationships, resting membrane potentials, and express similar ionic channels (e.g., CaV1.2, BK channels, RyRs, etc) (PMID: 28119464, PMID: 29070899, PMID: 23232643). In response to the reviewer, we have modified the discussion section of the manuscript to specifically note the mouse is not identical to the human but does share some common important features that make mice a good approximate model.

**Reviewer #2 (Public Review):**
In this study, Hernandez-Hernandez et al developed a gender-dependent mathematical model of arterial myocytes based on a previous model and new experimental data. The ionic currents of the model and its sex difference were formulated based on patch-clamp experimental data, and the model properties were compared with single-cell and tissue scale experimental results. This is a study that is of importance for the modeling field as well as for experimental physiology.

Thank you for the comment. In fact, we developed a model that incorporates sex-dependent differences that allowed for male and female models. It’s an important distinction as sex is a biological variable and gender is a self-ascribed characteristic.

**Reviewer #3 (Public Review):**
Summary:This hybrid experimental/computational study by Hernandez-Hernandez sheds new light on sex-specific differences between male and female arterial myocytes from resistance arteries. The authors conduct careful experiments in isolated myocytes from male and female mice to obtain the data needed to parameterize sex-specific models of two important ionic currents (i.e., those mediated by CaV1.2 and KV2.1). Available experimental data suggest that KV1.5 channel currents from male and female myocytes are similar, but simulations conducted in the novel Hernandez-Hernandez sex-specific models provide a more nuanced view. This gives rise to the first of the authors' three key scientific claims: (1) In males, KV1.5 is the dominant current regulating membrane potential; whereas, in females, KV2.1 plays a primary role in voltage regulation. They further show that this (2) the latter distinction drives drive sex-specific differences in intracellular Ca2+ and cellular excitability. Finally, working with one-dimensional models comprising several copies of the male/female myocyte models linked by resistive junctions, they use simulations to (3) predict that the sensitivity of arterial smooth muscle to Ca2+ channel-blocking drugs commonly used to treat hypertension is heightened in female compared to male cells.Strengths:The Methodology is described in exquisite detail in straightforward language that will be easy to understand for most if not all peer groups working in computational physiology. The authors have deployed standard protocols (e.g., parameter fitting as described by Kernik et al., sensitivity analysis as described by Sobie et al.) and appropriate brief explanations of these techniques are provided. The manoeuvre used to represent stochastic effects on voltage dynamics is particularly clever and something I have not personally encountered before. Collectively, these strengthen the credibility of the model and greatly enrich the manuscript.

We appreciate your comment highlighting the robustness of our methodology. Your acknowledgment of our approach to represent stochastic effects on voltage dynamics is especially encouraging. Indeed, noise is a fundamental component of physiological systems, including in vascular myocytes

Broadly speaking, the Results section describes findings that robustly support the three key scientific claims outlined in my summary. While there is certainly room for further discussion of some nuanced points as outlined below, it is evident these experiments were carefully designed and carried out with care and intentionality. In the present version of the manuscript, there are a few figures in which experimental data is shown side-by-side with outputs from the corresponding models. These are an excellent illustration of the power of the authors' novel sex-specific computational simulation platform. I think these figures will benefit from some modest additional quantitative analysis to substantiate the similarities between experimental and computational data, but there is already clear evidence of a good match.

We sincerely appreciate your constructive feedback on the Results section. We have included additional quantitative analysis to substantiate the similarities between experimental and computational data. We agree with the reviewer that the suggestion on the potential value of a more quantitative assessment. As such we have updated the figure to include an in-depth analysis that provides greater insights and solidifies the power of our simulation predictions when compared to experimental results. A detailed analysis of the male and female data as well as the male and female simulations are summarized in the text as follows:

Baseline membrane potential is -40 mV in male myocytes compared to -30 mV. The frequency of hyperpolarization transients (THs) is 1 Hz in male and 2.5 Hz in female cells for the specific baseline membrane potential shown in Figure 5 A-B. In the range of membrane potentials from -50 mV to -30 mV the frequency increases from 1-2.8Hz which is identical to the experimental frequency range.

Areas for Improvement:The authors used experimental data from a prior publication to calibrate their model of the BKCa current. As indicated in the manuscript, these data are for channel activity measured in a heterologous expression system (*Xenopus* oocytes). A similar principle applies to other major ion channels/pumps/etc. Is it possible there might be relevant sex-specific differences in these players as well? In the context of the present work, this feels like an important potential caveat to highlight, in case male/female differences in the activity of BKCa or other currents might influence model-predicted differences (e.g., the relative importance of KV1.5 and KV2.1). This should be discussed, and, if possible, related to the elegant sensitivity analysis presented in Fig. 5C (which shows, for example, that the models are relatively insensitive to variation in GBK).

We fully agree with the reviewer - an important caveat to highlight is the unknown sex-specific differences in all the other players regulating membrane potential and calcium signaling. While our initial assessments indicated that the contribution of BKCa channels to the total voltage-gated K+ current (IKvTOT) was small within the physiological range of -50 mV to -30 mV, further analysis of spontaneous transient outward currents revealed sex-specific variations. We have investigations underway to explore if BKCa channel expression and organization may be also sex-dependent.

The authors state that their model can be expanded to 2D/3D applications, "transitioning seamlessly from single-cell to tissue-level simulations". I would like to see more discussion of this. For example, given the modest complexity of the cell-scale model, how considerable would the computational burden be to implement a large network model of a subset of the human female or male arterial system? Are there sex-specific differences in vessel and/or network macro-structure that would need to be considered? How would this influence feasibility? Rather than a 1D cable as implemented here, I imagine a multi-scale implementation would involve the representation of myocytes wrapped around vessels. How would the behavior of such a system differ from the authors' presented work using a 1D representation of 100 myocytes coupled end-to-end? Could these differences partially explain why the traces in Fig. 8D are smoother than those in Fig. 8C? From my standpoint, discussing these points would enrich the paper.

We appreciate the reviewer’s thoughtful and forward-looking ideas! Indeed, we are very interested to extend the model to incorporate a number of these important items.

Our choice for the 1D cable model was driven by its anatomical relevance to the structure of third and fourth-order mesenteric arteries. These arteries possess a singular layer of vascular myocytes encircling the lumen in a cylindrical arrangement. When we conceptualize this structure as unrolled or viewed laterally, it aligns with a flat, rectangular form, closely paralleling our 1D cable implementation. One option is to expand this into a 2D representation by connecting multiple 1D cables together. Another option would be to connect the 1D cable end-to-end to create a ring to represent a cross section. While these approaches would appear to be different geometries, in either case, the dynamics will remain consistent because the cells comprising the tissue are the same. There is no propagating impulse (for example – although even then in a 2D homogenous tissue, a planar wave is identical in 1D), and the only effect will be an increase in electrotonic load (sink) from neighboring cells, which can readily be approximated in 1D by increasing coupling or modification of the boundary conditions.

We totally agree that future investigation should include exploration into the potential sex-specific differences in vessel and/or network macro-structure, as these factors may critically impact predictions and indeed the difference in traces observed between Fig. 8D and Fig. 8C may well involve “insulating” effects of vessel layers and interaction between various cell types and other structural factors. In particular, the contribution of endothelial cells in modulating membrane potential in vascular myocytes might be one such influential factor. In future studies, we are also keen to investigate blood flow regulation where a 3D configuration might become necessary.

The nifedipine data presented in Fig. 9 are quite compelling, and a nice demonstration of the potential power of the new models. How does this relate to what is known about the clinical male/female responses to nifedipine? Are there sex differences in drug efficacy?

Thank you for your comment regarding Fig. 9.

It is well known that sex-specific differences in pharmacokinetics and pharmacodynamics influence antihypertensive drug responses [PMID: 8651122., PMID: 22089536]. Previous studies, notably by Kloner et al., have illustrated this point quantitatively, highlighting a more pronounced diastolic BP response in women (91.4%) compared to men (83%) when treated with dihydropyridine-type channel blockers, such as amlodipine/nifedipine. Importantly, this distinction persisted even after adjusting for confounding factors such as baseline BP, age, weight, and dosage per kilogram [PMID: 8651122]. An interesting observation from Kajiwara et al. emphasizes that vasodilation-related adverse symptoms occur significantly more frequently in younger women (<50 years) compared to their male counterparts, suggesting a heightened sensitivity to dihydropyridine-type calcium channel blockers [PMID: 24728902].

While our findings resonate with clinical observations, a word of caution is in order. Our data suggest that, in the mouse model, nifedipine elicits distinct sex-specific effects. Importantly, future research should test the direct translatability and implications of these observations in human subjects.

**Reviewer #1 (Recommendations For The Authors):**
1. Cellular simulations with noise: It might be useful to also include in this section how noise was introduced specifically into the [Ca]SR equations.

We agree. The manuscript now includes an expanded explanation of how noise was incorporated into the model. This includes the addition of Equation 6 into section 2.4 "Cellular simulations with noise" to describe how noise was specifically integrated into the [Ca]SR equations. Please see LINE 355.

1. For equation 14, the description might be confusing. RCG and Ri are not explicitly included.

Thank you – this has been corrected.

1. In the paragraph starting with, "Having explored the regulation of graded membrane potential..." , the references to Figure 7C-D do not seem to match the content of the text. Namely, the figures show female versus male responses to nifedipine, which is not introduced until the next paragraph. Additionally, the graphs in 7C-D do not have the panels titled and the y-axes labeled.

We apologize for the error. We have modified the text and figures to address these issues.

1. Perhaps give more detail on how the effects of nifedipine were mathematically simulated at the ionic current level.

Good suggestion. Briefly, previous studies [PMID: 1329564] have shown that at the therapeutic dose of nifedipine (i.e., about 0.1 μM) L-type Cav1.2 channel currents are reduced by about 70%. Accordingly, we decreased ICaL in our mathematical simulations by the same extent. It is known that dihydropyridine-type channel blockers exhibit a voltage-dependent behavior, predominantly binding to the inactivated state. In smooth muscle cells, these blockers initiate inhibition quickly within a voltage range of -60 to -40 mV. This range aligns with the membrane potential baseline of vascular muscle cells (PMID: 8388295), ensuring the blockers are effective without the need of inducing significant depolarization. Therefore, the voltage dependency of dihydropyridine-type channel blockers can be neglected.

1. For the simulations with 400 uncoupled myocytes, the methods stated that the "gap junctional resistance [was set] to zero". Did the authors mean to use "conductivity" or am I misunderstanding?

Thank you for bringing up this issue with the term "gap junctional resistance." We now state that the "gap junctional conductivity" was set to zero to indicate no electrical communication/coupling.

1. Address whether there are differences-such as in cell geometry, degree of sex-based ionic current changes, and frequency of spontaneous hyperpolarization-between mice and human smooth muscle myocytes that could limit the predictive capability of the model.

Excellent point. Our model was parametrized using data from mouse mesenteric artery smooth muscle cells as initial proof of concept. In general terms, mouse arteries are a good animal model for human arteries, as they have similar intravascular pressure-myogenic tone relationships, resting membrane potentials, and express similar ionic channel (e.g., CaV1.2, BK channels, RyRs, etc) (PMID: 28119464, PMID: 29070899). Unfortunately, these studies have largely been done in male arteries and myocytes. Thus, while we recognize that the physiological distinctions between mice and humans could introduce variances in the model's outcomes. Our model offers valuable insights into the sex-specific mechanisms of KV2.1 and CaV1.2 channels in controlling membrane potential and Ca2+ dynamics in mice. It has been shown that sex-specific differences in pharmacokinetics and pharmacodynamics influence antihypertensive drug responses [PMID: 8651122., PMID: 22089536]. Previous studies, notably by Kloner et al., have illustrated this point quantitatively, highlighting a more pronounced diastolic BP response in women (91.4%) compared to men (83%) when treated with dihydropyridine-type channel blockers, such as amlodipine/nifedipine. Importantly, this distinction persisted even after adjusting for confounding factors such as baseline BP, age, weight, and dosage per kilogram [PMID: 8651122]. An interesting observation from Kajiwara et al. emphasizes that vasodilation-related adverse symptoms occur significantly more frequently in younger women (<50 years) compared to their male counterparts, suggesting a heightened sensitivity to dihydropyridine-type calcium channel blockers [PMID: 24728902].

While our findings resonate with clinical observations, a word of caution is in order. Our data suggest that, in the mouse model, nifedipine elicits distinct sex-specific effects. Importantly, future research should test the direct translatability and implications of these observations in human subjects.

1. "A virtual drug-screening system that can model drug-channel interactions" (pg 32) sounds very novel.

Thank you for highlighting this. We recognize the typo in our manuscript and have made the necessary corrections to ensure clarity and accuracy.

**Reviewer #2 (Recommendations For The Authors):**
The manuscript is well written. I only have some minor comments:1. In the patch clamp experiments, there is no information on the recovery of the ionic currents. Is recovery important or not in arterial myocytes? This question is related to the results shown in Figs 5-7. In Fig.5, is the oscillation caused by noise alone or a spontaneous oscillation (such as the oscillation in Fis.6-7) modulated by noise? In general, recovery is an important parameter for the frequency of spontaneous oscillations. It seems to me that the spontaneous oscillations in Fig.8 are mainly noise-driven since they disappear after the cells are coupled through gap junctions.

One important aspect of the oscillatory behavior of the smooth muscle cells is the very long timescales, with fluctuations occurring on the order of seconds. But the majority of ion channels are operating and recovering on the order of milliseconds, so a reasonable approximation is that most ion channels in the cell are operating at steady state at low voltages.

Oscillations in Fig.5: Both the intrinsic oscillations and the noise play key roles in shaping in the oscillations.

The intrinsic deterministic dynamics of the model cells are oscillatory (as seen in Figures 6-7), but the noise can trigger sparks early or delay them, which leads to substantial fluctuations in the inter-spark intervals. Therefore, the spontaneous oscillations are technically modulated by the noise rather than driven by the noise. Nevertheless, in both cases, recovery dynamics play an essential role in shaping the oscillations and determining their frequency

Note however that, when an excitable system is around the bifurcation for oscillations and noise is included, the "firing" statistics in the oscillatory state and the non-oscillatory state are indistinguishable for moderate to high levels of noise.

Noise Exclusion in Figures 6-7: To offer a clear and undistracted interpretation of the results, noise was intentionally omitted from Figures 6-7. This was done to ensure that the primary phenomena under investigation were not obscured. While we recognize the significance of incorporating all elements, including noise, in simulating biological systems, in this case we prioritized a clear point to be made in this context.

Oscillations in Fig.8: Your observation regarding Fig.8 is insightful. Here, uncoupled cells indeed display a spontaneous oscillatory behavior. As documented in previous research, this behavior is not an artifact resulting from cell isolation from the vessel but represents an intrinsic characteristic vital for maintaining electrical signals. The noise in the cells leads to substantial fluctuations in the inter-spike intervals. Because the noise in each cell is uncorrelated, it acts to desynchronize the activity of the cells. Therefore, instead of synchronizing the activity of the cells, the gap junction coupling quenches the large-scale oscillations (the spikes), creating lower amplitude irregular oscillations.

1. The calcium level is much higher in women than in men as shown in Figs.7 and 9. Do women have higher arterial pressure than men?

We thank the reviewer for the observation regarding the calcium levels in Figs.7 and 9. All data presented comes from both male and female C57BL/6J animal models, forming the foundation of our experimental framework.

From earlier studies by the Santana lab (PMID: 32015129), distinct sex-specific differences were found between male and female vascular mesenteric vessels. When the endothelium was removed from small arteriole segments and these segments were subsequently pressurized within a range of 20–120 mmHg, the female arterioles exhibited a pronounced myogenic response in comparison to the male ones. This brings to the forefront the marked sex-based differences, especially in the context of vascular smooth muscle activity.

Yet, when examining the behavior of whole, intact vessels, a different picture emerges. Despite clear sex-specific differences in conditions with the endothelium removed, these distinctions become less pronounced in whole, intact vessels. In essence, both male and female mice exhibit analogous arterial pressure patterns. This suggests possible compensatory mechanisms related to the caliber and structure of the small vessels.

To address the core issue: Despite our data showing higher calcium levels in female samples, it doesn't necessarily imply females consistently exhibit higher arterial pressure across all physiological scenarios.

1. In Fig.9, where is the intravascular pressure (a variable or a parameter) in the mathematical model?

In our model, the intravascular pressure effects are implicitly introduced by modulating the conductance of the non-selective cation currents (INSCC). Specifically, the increase in INSCC is our way of simulating the effects of pressure-induced membrane depolarization. This approach allows us to capture the physiological response to intravascular pressure changes without explicitly introducing it as a separate parameter in the model. We have modified the manuscript to ensure that this rationale is clarified.

1. In Eq.14, the given units of Rmyo (Ohmcm) and Rg (Ohmcmcm) are different, but Eq.14 implies they should have the same unit.

We sincerely appreciate the reviewer's meticulous observation regarding the units discrepancy in Eq.14. We have revised the manuscript to correct the error.

**Reviewer #3 (Recommendations For The Authors):**
Suggestions for improved or additional experiments, data, or analyses:Fig. 5 A-B: This is a beautiful qualitative comparison between experimental and simulation data! I think it would be even more impactful if the authors carried out some quantitative analysis of the similarity between male/female experimental/simulation data. For example, the "resting" Vm levels (approx. -30 mV and -40 mV for females and males, respectively) and the peak levels of Vm hyperpolarization could be compared, as well as the frequency of transient hyperpolarization events. It seems like the female model is much more prone to intervals of relative quiescence (i.e., absence of transient hyperpolarization events - e.g., from ~5-6.5 s). Is this consistent with the duration of such ranges in the experimental data (e.g., from 0 to 2.5 s in Fig. 5A).

Thank you for your positive remarks concerning the qualitative comparison in Fig. 5 A-B. We are indeed enthusiastic about the parallels we've identified between experimental and simulation outcomes. We agree with the reviewer that the suggestion on the potential value of a more quantitative assessment. As such we have updated the figure to include an in-depth analysis that provides greater insights and solidifies the power of our simulation predictions when compared to experimental results. A detailed analysis of the male and female data as well as the male and female simulations are summarized in the text as follows:

Baseline membrane potential is -40 mV in male myocytes compared to -30 mV. The frequency of hyperpolarization transients (THs) is 1 Hz in male and 2.5 Hz in female cells for the specific baseline membrane potential shown in Figure 5 A-B. In the range of membrane potentials from -50 mV to -30 mV the frequency increases from 1-2.8Hz which is identical to the experimental frequency range.

• Fig. 7 C-D: Likewise, it would be helpful to quantitatively characterize male/female differences in the model's response to simulated Ca channel blockade (e.g., rate of transient hyperpolarization events, relative levels of ICa and [Ca]i).

Thank you for the constructive feedback on Fig. 7 C-D. We appreciate the emphasis on a quantitative approach to solidify our understanding and have modified the results as follows:

Next, we simulated the effects of calcium channel blocker nifedipine on ICa at a steady membrane potential of -40 mV in male and female simulations. Briefly, previous studies70 have shown that at the therapeutic dose of nifedipine (i.e., about 0.1 μM) L-type Cav1.2 channel currents are reduced by about 70%. Accordingly, we decreased ICa in our mathematical simulations by the same extent. In Figure 7C-D, we show the predicted male (gray) and female (pink) time course of membrane voltage at -40 mV (top panel), ICa (middle panel), and [Ca2+]i (lower panel). First, we observed that in both male and females 0.1 μM nifedipine modifies the frequency of oscillation in the membrane potential, by causing a reduction in oscillation frequency. Second, both male and female simulations (middle panels) show that 0.1 μM nifedipine caused a reduction of ICa to levels that are very similar in male and female myocytes following treatment. Consequently, the reduction of ICa causes both male and female simulations to reach a very similar baseline [Ca2+]i of about 85 nM (lower panels). As a result, simulations provide evidence supporting the idea that CaV1.2 channels are the predominant regulators of intracellular [Ca2+] entry in the physiological range from -40 mV to -20 mV. Importantly, these predictions also suggest that clinically relevant concentrations of nifedipine cause larger overall reductions in Ca2+ influx in female than in male arterial myocytes.

Recommendations for improving the writing and presentation:When I accessed the GitHub repository linked in section 2.7 (Aug 17, 13:30 PT) it only contained a LICENSE file and none of the described codes and model equations appeared to be publicly available. I would like to access and examine these files. Based on the Clancy lab's excellent track record for making their work publicly available, I have no doubt that the published files will be complete, thoroughly documented, and ready for implementation in studies to reproduce or extend the work described in this manuscript.
https://github.com/ClancyLabUCD/sex-specific-responses-to-calcium-channel-blockers-in-mesenteric-vascular-smooth-muscle

We sincerely apologize for the omission regarding the GitHub repository. It was never our intention to omit the crucial files that should accompany our manuscript. We deeply regret any inconvenience this may have caused in your review process.

We deeply value transparency and the importance of making our work accessible to fellow researchers and the wider community. As you rightly pointed out, the Clancy lab has always been committed to ensuring that our work is available publicly, and this instance is no exception. Please find all codes and documentation here:

Minor corrections to the text and figures:The introduction is somewhat lengthy, and some of the material contained therein might be more suitable to be merged into the Discussion instead (e.g., paragraphs on negative feedback regulation and the recent study by O'Dwyer et al.).

Thank you – we have updated the introduction but left some foundational work descriptions intact.

• Page 6, section 1.1: There is a missing word (mice?) in the first sentence.• Page 11, under Eqn. 7: Luo is misspelled as Lou. (Also twice on Page 20.)

Thank you – these have been corrected.

Figs. 2-3: As a colorblind person, it was somewhat challenging for me to differentiate between the red and black lines. Choosing a higher-contrast colour pairing would be beneficial. For some reason, this is not so much of an issue for other figures that use the red/black scheme later in the manuscript (e.g., Figs. 5, 7-8).

We truly appreciate your feedback on the color contrast used in our figures. Accessibility and clarity are crucial to us, and we regret any difficulty you encountered due to the color choices. Based on your valuable feedback, we have included different color pairings in our visual representations to ensure they are comprehensible to all readers, including those who are colorblind.

Fig. 2-3: I am also confused about the use of symbols to indicate significant differences in these plots. In Fig. 2, ** is defined in the legend but not used in the figure. In both figures, the symbols are placed above/below specific sets of points, but it is unclear whether large differences for other x-axis values are statistically significant (e.g., -20 mV in Fig. 3B, +40 mV in Fig. 2C, etc.) This should be clarified.

Thank you – we now have included all the significant differences in the data discussed in the manuscript.

Page 22: The authors state that they "introduced noise into the [Ca]SR..." but the specifics of this approach are not described. As with other aspects of the Methods section, it would be suitable to provide a brief description of the technique used in ref. 40, perhaps added to section 2.4.

Thank you – it has been corrected.

Fig.7 C-D: Axis labels and units are missing. Even though the labels and units will be inferred by most readers, it would be helpful to include them here (at least in C).

Thank you for pointing out the inconsistency between the textual references and Figure 7C-D. We have added the corrected figure.

Page 32: "...the first step toward the development of a virtual drug-screaming system..." I think the authors mean drug-screening. As a side note, this is immediately in the running for the best typo I've ever seen as a peer reviewer.

Thank you for pointing out this error, and we sincerely appreciate your sense of humor about it. You are indeed correct; the intended word is "drug-screening." We have corrected this typo in the manuscript. We're grateful for your thorough review and the light-hearted way you brought this to our attention.